# Bridge the Inference Gaps of Neural Processes via Expectation Maximization

**Qi Wang**,[*] **Marco Federici, Herke van Hoof**
AMLab, University of Amsterdam, 1098XH, Amsterdam, the Netherlands
`hhq123go@gmail.com, m.federici@uva.nl, h.c.vanhoof@uva.nl`

## Abstract

The neural process (NP) is a family of computationally efficient models for learning distributions over functions. However, it suffers from under-fitting and shows suboptimal performance in practice. Researchers have primarily focused on incorporating diverse structural inductive biases, *e.g.* attention or convolution, in modeling. The topic of inference suboptimality and an analysis of the NP from the optimization objective perspective has hardly been studied in earlier work. To fix this issue, we propose a surrogate objective of the target log-likelihood of the meta dataset within the expectation maximization framework. The resulting model, referred to as the Self-normalized Importance weighted Neural Process (SI-NP), can learn a more accurate functional prior and has an improvement guarantee concerning the target log-likelihood. Experimental results show the competitive performance of SI-NP over other NPs objectives and illustrate that structural inductive biases, such as attention modules, can also augment our method to achieve SOTA performance.

## 1 Introduction

The combination of deep neural networks and stochastic processes provides a promising framework for modeling data points with correlations (Ghahramani, 2015). It exploits the high capacity of deep neural networks and enables uncertainty quantification for distributions over functions.

As an example, we can look at the deep Gaussian process (Damianou & Lawrence, 2013). However, the run-time complexity of predictive distributions in Gaussian processes is cubic *w.r.t.* the number of predicted data points. To circumvent this, Garnelo et al. (2018a;b) developed the family of neural processes (NPs) as the alternative, which can model more flexible function distributions and capture predictive uncertainty at a lower computational cost.

In this paper, we study the vanilla NP as a deep latent variable model and show the generative process in Fig. (1). In particular, let us recap the inference methods used in vanilla NPs: It learns to approximate the functional posterior $q_\phi(z) \approx p(z|\mathcal{D}^T; \vartheta)$ and a functional prior $q_\phi(z|\mathcal{D}^C) \approx p(z|\mathcal{D}^C; \vartheta)$, which are permutation invariant to the order of data points. Then the predictive distribution for a data point $[x_*, y_*]$ can be formulated in the form $\mathbb{E}_{q_\phi(z|\mathcal{D}^C)}[p(y_*|[x_*, z]; \vartheta)]$.

While the NP provides a computationally efficient framework for modeling exchangeable stochastic processes, it exhibits underfitting and fails to capture accurate uncertainty (Garnelo et al., 2018b; Kim et al., 2019) in practice. To improve its generalization capability, researchers have focused much attention on finding appropriate inductive biases, *e.g.* attention (Kim

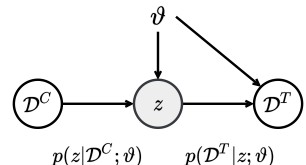

Figure 1: **Deep Latent Variable Models for Neural Processes.** Here $\mathcal{D}^C$ and $\mathcal{D}^T$ respectively denote the context points for the functional prior inference and the target points for the function prediction. The global latent variable $z$ is to summarize function properties. The model involves a functional prior distribution $p(z|\mathcal{D}^C; \vartheta)$ and a functional generative distribution $p(\mathcal{D}^T|z; \vartheta)$. Please refer to Section (2) for detailed notation descriptions.

---

[*]Correspondence Author.

et al., 2019) and convolutional modules (Gordon et al., 2019; Kawano et al., 2020), Bayesian mixture structures (Wang & van Hoof, 2022) or Bayesian hierarchical structures (Naderiparizi et al., 2020), to incorporate in modeling.

**Research Motivations.** Most previous work (Garnelo et al., 2018a;b; Kim et al., 2019; Gordon et al., 2019; Wang & van Hoof, 2022) ignores the reason why the vanilla NP suffers the performance bottleneck and what kind of functional priors the vanilla NPs can represent. In particular, we point out the remaining crucial issues that have not been sufficiently investigated in this domain, respectively: (i) understanding the inference suboptimality of vanilla NPs (ii) quantifying statistical traits of learned functional priors. To this end, we try to diagnose the vanilla NP from its optimization objective. Our primary interest is to find a tractable way to optimize NPs and examine the statistics of learned functional priors from diverse optimization objectives.

**Developed Methods.** To understand the inference suboptimality of vanilla NPs, we establish connections among a collection of optimization objectives, *e.g.* approximate evidence lower bounds (ELBOs) and Monte Carlo estimates of log-likelihoods, in Section (3). Then we formulate a tractable optimization objective within the variational expectation maximization framework and obtain the Self-normalized Importance weighted neural process (SI-NP) in Section (4).

**Contributions.** To summarize, our primary contributions are three-fold: (i) we analyze the inherent inference sub-optimality of NPs from an optimization objective optimization perspective; (ii) we demonstrate the equivalence of conditional NPs (Garnelo et al., 2018a) and SI-NPs with one Monte Carlo sample estimate, which closely relates to the prior collapse in **Definition** (3.1); (iii) our developed SI-NPs have an improvement guarantee to the likelihood of meta dataset in optimization and show a significant advantage over baselines with other objectives.

## 2 PRELIMINARIES

**General Notations.** We study NPs in a meta learning setup. $\mathcal{T}$ defines a set of tasks with $\tau$ a sampled task. Let $\mathcal{D}_\tau^C = \{(x_i, y_i)\}_{i=1}^n$ and $\mathcal{D}_\tau^T = \{(x_i, y_i)\}_{i=1}^{n+m}$ denote the context points for the functional prior inference and the target points for the function prediction. The latent variable $z$ is a functional representation of a task $\tau$ with observed data points.

We refer to $\vartheta \in \Theta$ as the parameters of the deep latent variable model for NPs. In detail, $\vartheta$ consists of encoder parameters in a functional prior $p(z|\mathcal{D}_\tau^C; \vartheta)$ and decoder parameters in a generative distribution $p(\mathcal{D}_\tau^T|z; \vartheta)$. $\phi$ refer to the parameters of a variational posterior distribution $q_\phi(z) = q_\phi(z|\mathcal{D}_\tau^T)$, while $\eta$ refer to the parameters of a proposal distribution $q_\eta(z)$ in the following self-normalized importance sampling. Gaussian distributions with diagonal covariance matrices are the default choice for these distributions, *e.g.* $p(z|\mathcal{D}_\tau^C; \vartheta) = \mathcal{N}(z; \mu_\vartheta(\mathcal{D}_\tau^C), \Sigma_\vartheta(\mathcal{D}_\tau^C))$, $q_\phi(z) = \mathcal{N}(z; \mu_\phi(\mathcal{D}_\tau^T), \Sigma_\phi(\mathcal{D}_\tau^T))$ and $q_\eta(z|\mathcal{D}_\tau^T) = \mathcal{N}(z; \mu_\eta(\mathcal{D}_\tau^T), \Sigma_\eta(\mathcal{D}_\tau^T))$.

**NPs as Exchangeable Stochastic Processes.** In vanilla NPs, the element-wise generative process can be translated into Eq. (1). Here the mean and variance functions are respectively denoted by $\mu$ and $\Sigma$.

$$\rho_{x_{1:n+m}}(y_{1:n+m}) = \int p(z) \prod_{i=1}^{n+m} \mathcal{N}(y_i; \mu(x_i, z), \Sigma(x_i, z)) dz \tag{1}$$

Based on the Kolmogorov extension theorem (Klenke, 2013) and de Finneti's theorem (Kerns & Székely, 2006), the above equation $\rho_{x_{1:n+m}}(y_{1:n+m})$ is verified to be a well-defined exchangeable stochastic process.

**NPs in Meta Learning Tasks.** Given a collection of tasks $\mathcal{T}$, we can decompose the marginal distribution $p(\mathcal{D}_\mathcal{T}^T|\mathcal{D}_\mathcal{T}^C; \vartheta)$ with a global latent variable $z$ in Eq. (2). Here the conditional distribution $p(z|\mathcal{D}_\tau^C; \vartheta)$ with $\tau \in \mathcal{T}$ is permutation invariant *w.r.t.* the order of data points and encodes the functional prior in the generative process.

$$\underbrace{p(\mathcal{D}_\mathcal{T}^T|\mathcal{D}_\mathcal{T}^C; \vartheta)}_{\text{Marginal Likelihood}} = \prod_{\tau \in \mathcal{T}} \left[ \int \underbrace{p(\mathcal{D}_\tau^T|z; \vartheta)}_{\text{Generative Likelihood}} \underbrace{p(z|\mathcal{D}_\tau^C; \vartheta)}_{\text{Functional Prior}} dz \right] \tag{2}$$

Throughout the paper, the optimization objective of our interest is the marginal log-likelihood of a meta learning dataset in Eq. (3). Furthermore, this applies to all NP variants.

$$\max_{\vartheta} \sum_{\tau \in \mathcal{T}} \ln \left[ \int p(\mathcal{D}_\tau^T | z; \vartheta) p(z | \mathcal{D}_\tau^C; \vartheta) dz \right] \tag{3}$$

For the sake of simplicity, we consider one task $\tau$ to derive equations in the following section, which corresponds to maximizing the following objective[1].

$$\mathcal{L}(\vartheta) = \ln \left[ \int p(\mathcal{D}_\tau^T | z; \vartheta) p(z | \mathcal{D}_\tau^C; \vartheta) dz \right] \tag{4}$$

In the NP family (Garnelo et al., 2018a;b), the target data points are conditionally independent given the global latent variable $p(\mathcal{D}_\tau^T | z; \vartheta) = \prod_{i=1}^{n+m} p(y_i | [x_i, z]; \vartheta)$. The marginal distribution can be interpreted as the infinite mixture of distributions when $z$ is defined on a continuous domain. Now learning distributions over functions is reduced to a probabilistic inference problem.

## 3 Optimization Gaps and Statistical Traits

### 3.1 Inference Suboptimality in vanilla NPs

Previously, variational auto-encoder (VAE) models (Kingma & Welling, 2013; Rezende et al., 2014) mostly set a prior distribution fixed, *e.g.* $\mathcal{N}(0, I)$, as the default to approximate the posterior. This differs significantly from NPs family settings. On the one hand, the functional prior is learned in NPs. On the other hand, the functional prior participates in the performance evaluation.

**Exact ELBO for NPs.** Following the essential variational inference operation, we can establish connections between the exact ELBO and the log-likelihood in Eq. (5). Given the functional prior $p(z | \mathcal{D}_\tau^C; \vartheta)$ and the generative distribution $p(\mathcal{D}_\tau^T | z; \vartheta)$, the exact functional posterior can be obtained by the Bayes rule

$$p(z | \mathcal{D}_\tau^T; \vartheta) = \frac{p(\mathcal{D}_\tau^T | z; \vartheta) p(z | \mathcal{D}_\tau^C; \vartheta)}{\int p(\mathcal{D}_\tau^T | z; \vartheta) p(z | \mathcal{D}_\tau^C; \vartheta) dz}.$$

The denominator $p(D_\tau^T | D_\tau^C)$ makes exact inference infeasible, and the family of variational posteriors is introduced to approximate $p(z | \mathcal{D}_\tau^T; \vartheta)$. Here the variational posterior family is defined in a parameterized set $\mathcal{Q}_\Phi = \{q_\phi(z) | \phi \in \Phi\}$.

$$\mathcal{L}(\vartheta) = \ln p(\mathcal{D}_\tau^T | \mathcal{D}_\tau^C; \vartheta) = \underbrace{\mathbb{E}_{q_\phi(z)} \left[ \ln \frac{p(\mathcal{D}_\tau^T, z | \mathcal{D}_\tau^C; \vartheta)}{q_\phi(z)} \right]}_{\text{Exact ELBO}} + \underbrace{D_{KL} \left[ q_\phi(z) \parallel p(z | \mathcal{D}_\tau^T; \vartheta) \right]}_{\text{Posterior Approximation Gap}} \tag{5}$$

When the variational posterior family is flexible enough, *e.g.* $p(z | \mathcal{D}_\tau^T; \vartheta) \in \mathcal{Q}_\Phi$, the posterior approximation gap can be reduced to an arbitrarily small quantity. In this case, maximizing the exact ELBO in Eq. (6) increases the likelihood in Eq. (5) accordingly.

$$\mathcal{L}_{\text{ELBO}}(\vartheta, \phi) = \mathbb{E}_{q_\phi(z)} \left[ \ln p(\mathcal{D}_\tau^T | z; \vartheta) \right] - D_{KL} \left[ q_\phi(z) \parallel p(z | \mathcal{D}_\tau^C; \vartheta) \right] \tag{6}$$

**Approximate ELBO for NPs.** As previously mentioned, the inference is complicated since the functional prior and the posterior in the exact ELBO are unknown. To this end, Garnelo et al. (2018b) proposes a surrogate objective as an approximate ELBO for NPs. This is defined as Eq. (7) to maximize.

$$\mathcal{L}_{\text{NP}}(\vartheta, \phi) = \mathbb{E}_{q_\phi(z)} \left[ \ln \underbrace{p(\mathcal{D}_\tau^T | z; \vartheta)}_{\text{Generative Likelihood}} \right] - \underbrace{D_{KL} \left[ q_\phi(z) \parallel q_\phi(z | \mathcal{D}_\tau^C) \right]}_{\text{Consistent Regularizer}} \tag{7}$$

The Kullback-Leibler divergence between the approximate posterior $q_\phi(z)$ and the approximate prior $q_\phi(z | \mathcal{D}_\tau^C)$ is referred to as the consistent regularizer in this paper. We claim that the consistent regularizer is the source of the inference suboptimality of vanilla NPs, and this is shown in Appendix (D.2) as the proof of Remark (1).

---

[1]Meta training and testing phases are implemented in a batch of tasks consistent with Eq. (2)/(3).

**Remark 1** *Eq. ([7](#)) is an invalid variational inference objective, and optimizing it cannot guarantee to find optimal or locally optimal solutions for the maximization over $\sum_{\tau \in \mathcal{T}} \ln p(\mathcal{D}_\tau^T | \mathcal{D}_\tau^C; \vartheta)$.*

**Other Available Objectives in NPs Family.** Now we turn to other tractable optimization objectives in NPs. These include conditional neural processes (CNPs) ([Garnelo et al.](#), [2018a](#)) and convolutional neural processes (ConvNPs) ([Foong et al.](#), [2020](#)) (or VERSA ([Gordon et al.](#), [2018](#))).

The CNP is also a typical model of the NPs family. The objective $\mathcal{L}_{\text{CNP}}(\vartheta)$ can be obtained when the functional prior collapses into a Dirac delta distribution with $\hat{z}$ a fixed real-value vector.

$$\mathcal{L}_{\text{CNP}}(\vartheta) = \mathbb{E}_{p(z|\mathcal{D}_\tau^C; \vartheta)} \left[ \ln p(\mathcal{D}_\tau^T | z; \vartheta) \right] \quad \text{with} \quad p(z|\mathcal{D}_\tau^C; \vartheta) = \delta(|z - \hat{z}|) \tag{8}$$

For the ConvNP, we do not focus on the convolutional structural inductive bias and concentrate more on the optimization objective itself. Its objective in Eq. ([9](#)) is a biased Monte Carlo estimate of Eq. ([3](#)), so we can maximize the log-likelihood of marginal distributions straightforwardly.

$$\mathcal{L}_{\text{ML-NP}}(\vartheta) = \ln \left[ \frac{1}{B} \sum_{b=1}^{B} \exp \left( \ln p(\mathcal{D}_\tau^T | z^{(b)}; \vartheta) \right) \right] \quad \text{with} \quad z^{(b)} \sim p(z|\mathcal{D}_\tau^C; \vartheta) \tag{9}$$

This is termed as Monte Carlo Maximum Likelihood $\mathcal{L}_{\text{ML-NP}}(\vartheta)$[2], and $B$ is the number of used Monte Carlo samples. Without the involvement of the consistent regularizer, both $\mathcal{L}_{\text{CNP}}(\vartheta)$ and $\mathcal{L}_{\text{ML-NP}}(\vartheta)$ will not encounter the approximation gap in practice.

## 3.2 EVALUATION CRITERIA & ASYMPTOTIC PERFORMANCE

As in ([Le et al.](#), [2018](#); [Foong et al.](#), [2020](#)), we take a multi-sample Monte Carlo method to evaluate the performance. This applies to both meta training and meta testing processes. In detail, NP models need to run $B$ times stochastic forward pass by sampling $z^{(b)} \sim p(z|\mathcal{D}_\tau^C; \vartheta)$ and then compute the log-likelihoods as $\ln \left[ \frac{1}{B} \sum_{b=1}^{B} p(\mathcal{D}_\tau^T | z^{(b)}; \vartheta) \right]$.

After describing the evaluation criteria, we turn to a phenomenon of our interest. In Gaussian processes ([Ghahramani](#), [2015](#)), with more observed context points, the epistemic uncertainty can be decreased, and the predictive mean function is closer to the ground truth.

Similarly, this trait is also reflected in NPs family and can be quantitatively described as follows. Given a measure of average predictive errors $\beta$, the number of context points $n$ and the evaluated dataset, $\mathcal{D}_\tau^T$, the introduced metrics $\beta(\mathcal{D}_\tau^T; n)$ are decreased when increasing $n$ in prediction. In our paper, we refer to this trait as the *asymptotic behavior*.

**Definition 3.1 (Prior Collapse)** *The functional prior $p(z|\mathcal{D}_\tau^C; \vartheta) = \mathcal{N}(z; \mu_\vartheta(\mathcal{D}_\tau^C), \Sigma_\vartheta(\mathcal{D}_\tau^C))$ is said to collapse in learning when the trace of the covariance matrix satisfies $Tr[\Sigma_\vartheta(\mathcal{D}_\tau^C)] = \sum_{i=1}^{d} \sigma_i^2 \approx 0$ with $\Sigma_\vartheta(\mathcal{D}_\tau^C) = diag[\sigma_1^2, \ldots, \sigma_d^2]$.*

As previously mentioned, we can more precisely keep track of measures $\beta(\mathcal{D}_\tau^T; n)$, such as predictive log-likelihoods or mean square errors of data points, to assess the asymptotic behavior. The role of latent variables $z$ is to propagate the uncertainty about the partial observations in functions. And **Definition** ([3.1](#)) provides a quantitative way to examine the extent of prior collapse in ML-NPs and SI-NPs.

## 4 TRACTABLE OPTIMIZATION VIA EXPECTATION MAXIMIZATION

In this section, we propose alleviating the inference suboptimality of NPs with the help of the variational expectation maximization algorithm. The strategy is to formulate a surrogate optimization objective and then execute the EM-steps in optimization. The benefit of our method is to guarantee performance improvement *w.r.t.* the likelihood of meta dataset in iterations and finally result in at least a local optimum.

---

[2]The Monte Carlo maximum likelihood corresponds to the optimization objective that in ([Foong et al.](#), [2020](#)) except that the convolutional inductive bias is removed.

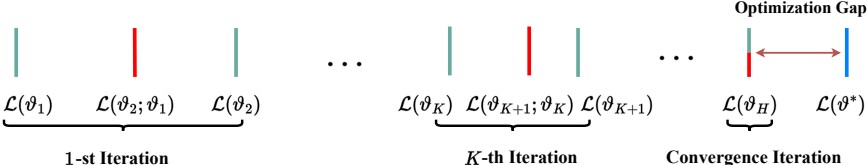

Figure 2: **Illustration of Expectation Maximization for NPs.** Green lines indicate the results after the E-steps while the red lines are for the M-steps in Algorithm (1). In the convergence iteration, the performance gap $\mathcal{L}(\vartheta_H) - \mathcal{L}(\vartheta_{H-1})$ is close to zero and the algorithm results in at least a local optimal solution. Values of these quantities are increased from the left to the right.

### 4.1 VARIATIONAL EXPECTATION MAXIMIZATION FOR NPS

This part is to avoid the inference suboptimality of vanilla NPs as mentioned earlier. We retain the neural architectures used in NPs. The basic idea is illustrated in Fig. (2). In detail, we iteratively construct the lower bound $\mathcal{L}(\vartheta_K)$ and maximize the surrogate function $\mathcal{L}(\vartheta; \vartheta_K)$. The referred optimization gap is due to the complexity of objectives or the choice of optimizers and measures the difference between converged (local) optimal functional prior and the theoretical optimal functional prior. The general pseudo code is Algorithm (1).

---

**Algorithm 1:** Variational Expectation Maximization for NPs.

---

**Input** : Task distribution $p(\mathcal{T})$; Task batch size, Number of particles, Initialized $\vartheta$ and $\eta$.
**Output:** Meta-trained parameters $\vartheta$ and $\eta$.

1 **for** $k = 1$ **to** $K$ **do**
2    E-step #1: $k \leftarrow k + 1$ and reset the variational posterior $q_\phi(z) = p(z|\mathcal{D}_\tau^T; \vartheta_k)$ in Eq. (5);
3    **if** *Use the Functional Prior as the Proposal* **then**
4       Reset $q_\eta(z|\mathcal{D}_\tau^T) = p(z|\mathcal{D}_\tau^C; \vartheta_k)$;
5    **else**
6       E-step #2: update the proposal $\eta_k = \arg\min_\eta \mathcal{L}_{\text{KL}}(\eta; \eta_{k-1}, \vartheta_k)$ in Eq. (29) according to operations in Appendix (E.3.1);
7    **end**
8    M-step: optimize surrogate functions $\vartheta_{k+1} = \arg\max_\vartheta \mathcal{L}_{\text{SI-NP}}(\vartheta; \eta_k, \vartheta_k)$ in Eq. (12);
9 **end**

---

#### 4.1.1 SURROGATE FUNCTION FOR EXACT NPS

To make the optimization of meta dataset log-likelihood feasible, we construct surrogate functions as the proxy in each iteration step. These meta learning surrogate functions with special properties are closely connected with the original objective, and we leave this discussion in Appendix (E).

Here $\vartheta_k$ denotes the parameter of the latent variable model for NPs in the $k$-th iteration of variational expectation maximization. Following the Algorithm (1), we take the E-step #1 by replacing the approximate posterior in Eq. (6) with the last time updated $p(z|\mathcal{D}_\tau^T; \vartheta_k)$. And this results in the following equation,

$$\mathcal{L}(\vartheta; \vartheta_k) = \mathbb{E}_{p(z|\mathcal{D}_\tau^T; \vartheta_k)} \left[ \ln p(\mathcal{D}_\tau^T, z|\mathcal{D}_\tau^C; \vartheta) - \ln p(z|\mathcal{D}_\tau^T; \vartheta_k) \right] \tag{10}$$

where $p(z|\mathcal{D}_\tau^T; \vartheta_k)$ is the posterior distribution.

**Proposition 1** *The proposed meta learning function $\mathcal{L}(\vartheta; \vartheta_k)$ in Eq. (10) is a surrogate function w.r.t. the log-likelihood of the meta learning dataset.*

The above proposition is examined based on the definition in Appendix (E.1).

#### 4.1.2 TRACTABLE OPTIMIZATION WITH SELF-NORMALIZED IMPORTANCE SAMPLING

Since the second term in Eq. (10) is constant in the iteration, we can drop it to simplify the surrogate objective as the right side of the following equation.

$$\max_\vartheta \mathcal{L}(\vartheta; \vartheta_k) \Leftrightarrow \max_\vartheta \mathcal{L}_{\text{EM}}(\vartheta; \vartheta_k) = \mathbb{E}_{p(z|\mathcal{D}_\tau^T; \vartheta_k)} \left[ \ln p(\mathcal{D}_\tau^T, z|\mathcal{D}_\tau^C; \vartheta) \right] \tag{11}$$

**Proposition 2** *Optimizing this surrogate function of a batch of tasks via the variational expectation maximization leads to an improvement guarantee w.r.t. the log-likelihood $\sum_{\tau \in \mathcal{T}} \ln p(\mathcal{D}_\tau^T | \mathcal{D}_\tau^C; \vartheta)$.*

Still we cannot optimize $\mathcal{L}_{\text{EM}}(\vartheta; \vartheta_k)$ since the expectation has no analytical solution and it is intractable to sample from $p(z|\mathcal{D}_\tau^T; \vartheta_k)$ for Monte Carlo estimates[3]. Remember that the marginal distribution $p(\mathcal{D}_\tau^T | \mathcal{D}_\tau^C; \vartheta_k)$ is task dependent and can not be ignored in computing the posterior $p(z|\mathcal{D}_\tau^T; \vartheta_k)$. To circumvent this, we introduce a proposal distribution $q_\eta(z|\mathcal{D}_\tau^T)$ and optimize the objective via self-normalized importance sampling (Tokdar & Kass, 2010). The resulting meta learning surrogate function is as follows:

$$\mathcal{L}_{\text{EM}}(\vartheta; \vartheta_k) = \mathbb{E}_{q_\eta} \left[ \frac{p(z|\mathcal{D}_\tau^T; \vartheta_k)}{q_\eta(z|\mathcal{D}_\tau^T)} \ln p(\mathcal{D}_\tau^T, z|\mathcal{D}_\tau^C; \vartheta) \right] \approx \sum_{b=1}^{B} \hat{\omega}^{(b)} \ln p(\mathcal{D}_\tau^T, z^{(b)}|\mathcal{D}_\tau^C; \vartheta)$$

$$= \sum_{b=1}^{B} \underbrace{\hat{\omega}^{(b)}}_{\text{Importance Weight}} \left[ \ln \underbrace{p(\mathcal{D}_\tau^T | z^{(b)}; \vartheta)}_{\text{Generative Likelihood}} + \ln \underbrace{p(z^{(b)} | \mathcal{D}_\tau^C; \vartheta)}_{\text{Functional Prior Likelihood}} \right] = \mathcal{L}_{\text{SI-NP}}(\vartheta; \eta_k, \vartheta_k) \tag{12}$$

where $z^{(b)} \sim q_{\eta_k}(z|\mathcal{D}_\tau^T)$, $\omega^{(b)} = \exp\left( \ln p(\mathcal{D}_\tau^T | z^{(b)}; \vartheta_k) + \ln p(z^{(b)} | \mathcal{D}_\tau^C; \vartheta_k) - \ln q_{\eta_k}(z^{(b)} | \mathcal{D}_\tau^T) \right)$ and $\hat{\omega}^{(b)} = \frac{\omega^{(b)}}{\sum_{b'=1}^{B} \omega^{(b')}}$.

In terms of the first conditional term in Eq. (12), all the data points are conditional independent and this is further expressed as $\ln p(\mathcal{D}_\tau^T | z^{(b)}; \vartheta) = \sum_{i=1}^{n+m} \ln p(y_i | [x_i, z^{(b)}]; \vartheta)$. In practice, the selection of proposal distributions is empirically tricky, so we make the update of proposal distributions optional in implementations. In our experimental settings, we simply use the functional prior $p(z|\mathcal{D}_\tau^C; \vartheta)$ as the default proposal distribution, which is competitive enough in performance.

**Proposition 3** *With one Monte Carlo sample used in Eq. (12), the presumed diagonal Gaussian prior $p(z|\mathcal{D}_\tau^C; \vartheta)$ will collapse into a Dirac delta distribution in convergence. In this case, SI-NP with the one sample Monte Carlo estimate in Eq. (13) is equivalent with CNP in Eq. (8).*

$$\mathcal{L}_{\text{SI-NP}}(\vartheta; \eta_k, \vartheta_k) \approx \mathbb{E}_{p(z|\mathcal{D}_\tau^C; \vartheta_k)} \left[ \ln \underbrace{p(\mathcal{D}_\tau^T | z; \vartheta)}_{\text{Generative Likelihood}} \right] + \underbrace{\mathbb{E}_{p(z|\mathcal{D}_\tau^C; \vartheta_k)} \left[ \ln p(z|\mathcal{D}_\tau^C; \vartheta) \right]}_{\text{Prior Collapse Term}} \tag{13}$$

The **Proposition** (3) establishes connections between SI-NPs and CNPs in optimization and we attribute this to the collapse term in Eq. (13). Hence, CNP can be viewed as a particular example in SI-NPs. The complete proof is based on limit analysis and can be found in Appendix (F.1).

## 4.2 SCALABLE TRAINING AND TESTING

As shown in **Algorithm** (1), the meta training process consists of two steps. We skip E-step #2 to avoid unstable optimization observed in empirical results. By repeating E-step #1/M-step iterations until convergence, the method can theoretically find at least a local optimal *w.r.t.* the log-likelihood of meta learning dataset based on the **Proposition** (2).

Once the learning progress reaches the final convergence, we can make use of the learned functional prior to obtain the predictive distribution. With $B$ particles in prediction, the distribution can be expressed as follows.

$$p(y|x, \mathcal{D}_\tau^C; \vartheta) = \mathbb{E}_{p(z|\mathcal{D}_\tau^C; \vartheta)} p(y|[x, z]; \vartheta) \approx \frac{1}{B} \sum_{b=1}^{B} p(y|[x, z^{(b)}]; \vartheta) \text{ with } z^{(b)} \sim p(z|\mathcal{D}_\tau^C; \vartheta) \tag{14}$$

---

[3]Though the exact posterior distribution $p(z|\mathcal{D}^T; \vartheta)$ can be inferred by Bayes rule, the denominator $\int p(\mathcal{D}_\tau^T | z; \vartheta) p(z|\mathcal{D}_\tau^C; \vartheta) dz$ is not available.

## 5 EXPERIMENTS AND ANALYSIS

In this section, two central questions are answered: (i) can variational EM based models SI-NPs, achieve a better local optimum than vanilla NPs? (ii) what is the role of randomness in functional priors? Specifically, we examine the influence of NPs optimization objectives on typical downstream tasks and understand the functional prior quantitatively.

**Baselines & Evaluations.** Since our concentration is on optimization objectives in NPs family, we compare to NP (Garnelo et al., 2018b), and CNP (Garnelo et al., 2018a), ML-NP (Foong et al., 2020) in experiments. Note that our developed SI-NP and ML-NP (Foong et al., 2020) are importance weighted models, but the mechanisms in estimating weights are significantly different. As for evaluations, we refer the reader to Sec. (3.2) for more information. Due to the page limit, additional experimental results are attached in Appendix (H.5), which shows that SI-NPs can achieve SOTA performance by adding attention networks.

### 5.1 SYNTHETIC REGRESSION

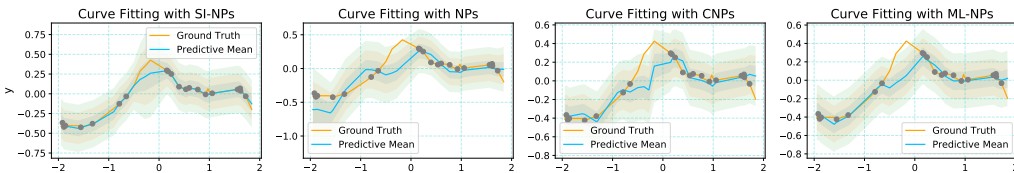

Figure 3: **Examples of Curve Fitting in RBF Kernel Cases.** The plots report predictive mean functions with $\pm 3$ standard deviations.

We create synthetic regression tasks by sampling functions from Gaussian processes. Three types of kernels, respectively Matern$-\frac{5}{2}$, RBF, and Periodic, are used to generate diverse function distributions. In each iteration, a batch of data points from functions is randomly processed into the context and the target for training models.

In meta testing, the same strategy is used to generate functions and data points. We report the average log-likelihood results in Table (1). It shows SI-NPs outperform the vanilla NPs in all kernel cases and are at least competitive with other baselines. ML-NPs are also superior to NPs, but they cannot beat CNPs in RBF and Periodic cases. An illustration of fitted curves is given in Fig. (3): compared to vanilla NPs, we notice that SI-NPs and ML-NPs can better match the ground truth and capture the uncertainty between context data points.

Table 1: Test average log-likelihoods of target data points for 1-dimensional Gaussian process dataset with various kernels (reported standard deviations in 5 runs). For each run, we randomly sample 1000 functions as tasks to evaluate.

| # | Matern $-\frac{5}{2}$ | RBF | Periodic |
|---|---|---|---|
| $\mathcal{L}_{\text{GP}}$ (Oracle) | $0.821_{\pm 0.03}$ | $1.18_{\pm 0.013}$ | $0.833_{\pm 0.017}$ |
| $\mathcal{L}_{\text{NP}}$ (Garnelo et al., 2018b) | $-0.225_{\pm 0.03}$ | $-0.183_{\pm 0.03}$ | $-0.611_{\pm 0.034}$ |
| $\mathcal{L}_{\text{CNP}}$ (Garnelo et al., 2018a) | $0.295_{\pm 0.017}$ | $0.463_{\pm 0.023}$ | $-0.533_{\pm 0.009}$ |
| $\mathcal{L}_{\text{ML-NP}}$ (Foong et al., 2020) | $0.303_{\pm 0.013}$ | $0.439_{\pm 0.009}$ | $-0.547_{\pm 0.036}$ |
| $\mathcal{L}_{\text{SI-NP}}$ (ours) | $0.305_{\pm 0.006}$ | $0.493_{\pm 0.007}$ | $-0.532_{\pm 0.036}$ |

### 5.2 IMAGE COMPLETION

Similar to experiments in (Garnelo et al., 2018b; Kim et al., 2019), we perform image completion experiments in this section. We implement NPs baselines in four commonly used datasets, namely MNIST (Bottou et al., 1994), FMNIST (Xiao et al., 2017), CIFAR10 (Krizhevsky et al., 2009) and SVHN (Sermanet et al., 2012). During the meta training, the goal is to complete images in which some pixels have been randomly masked out. More precisely, we learn a latent variable model that maps all pixel locations $x \in [0,1]^2$ to Gaussian distribution parameters of pixel values based on the context pixel locations and values. Fig. (4) is an example of completion results from sampled images. For implementation details, please refer to Appendix (G).

Table 2: Test average log-likelihoods with reported standard deviations for image completion in MNIST/FMNIST/SVHN/CIFAR10 (5 runs). We test the performance of different optimization objectives in both context data points and target data points. Except CNPs, we use 32 Monte Carlo samples from the functional prior to evaluate the average log-likelihoods.

| # | MNIST | | FMNIST | | SVHN | | CIFAR10 | |
|---|---|---|---|---|---|---|---|---|
| | context | target | context | target | context | target | context | target |
| $\mathcal{L}_{\text{NP}}$ | $0.81_{\pm 0.006}$ | $0.73_{\pm 0.007}$ | $0.83_{\pm 0.007}$ | $0.73_{\pm 0.009}$ | $3.19_{\pm 0.02}$ | $3.07_{\pm 0.02}$ | $2.35_{\pm 0.04}$ | $2.03_{\pm 0.02}$ |
| $\mathcal{L}_{\text{CNP}}$ | $1.05_{\pm 0.005}$ | $0.99_{\pm 0.008}$ | $0.95_{\pm 0.007}$ | $0.90_{\pm 0.009}$ | $\mathbf{3.57}_{\pm 0.003}$ | $3.48_{\pm 0.004}$ | $2.71_{\pm 0.004}$ | $2.53_{\pm 0.006}$ |
| $\mathcal{L}_{\text{ML-NP}}$ | $1.06_{\pm 0.004}$ | $0.99_{\pm 0.006}$ | $0.94_{\pm 0.008}$ | $0.89_{\pm 0.007}$ | $3.51_{\pm 0.008}$ | $3.43_{\pm 0.006}$ | $2.60_{\pm 0.005}$ | $2.41_{\pm 0.005}$ |
| $\mathcal{L}_{\text{SI-NP}}$ (ours) | $\mathbf{1.09}_{\pm 0.006}$ | $\mathbf{1.02}_{\pm 0.004}$ | $\mathbf{0.98}_{\pm 0.004}$ | $\mathbf{0.94}_{\pm 0.005}$ | $\mathbf{3.57}_{\pm 0.003}$ | $\mathbf{3.50}_{\pm 0.003}$ | $\mathbf{2.75}_{\pm 0.004}$ | $\mathbf{2.60}_{\pm 0.005}$ |

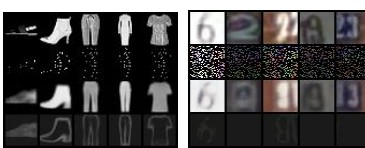

In the evaluation, the number of context pixels is randomly selected for each image in the dataset. We examine the performance in a testing set of the above image datasets and report the average log-likelihoods in Table (2). We can see that SI-NP achieves the best performance in all image datasets. The performance gaps between SI-NPs and NPs are remarkable. Furthermore, the asymptotic behaviors of all baselines are illustrated in Fig. (5). All models exhibit asymptotic behaviors by varying the number of context points, and NPs mainly result in clearly lower log-likelihoods with 10 context points. The observations with different number of context pixels are consistent with the conclusion in Table (2).

Figure 4: **SI-NP Completed Images.** From top to bottom in rows are original images, context points, learned predictive means and variances of sampled images.

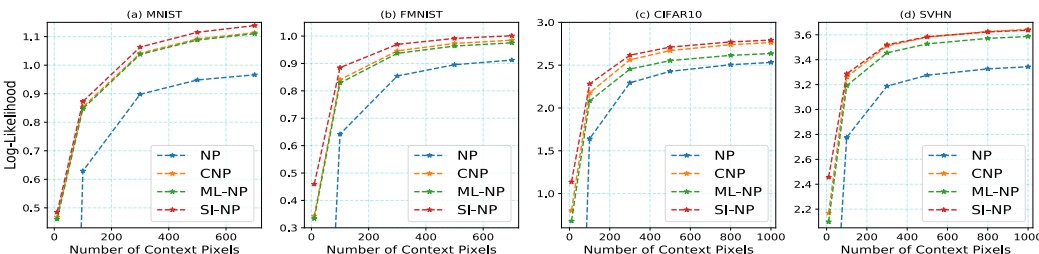

Figure 5: **Asymptotic Performance in Image Completion.** We meta test pixel average log-likelihoods with varying number of context points in image datasets. Context points are randomly selected for each image in testing processes. For MNIST/FMNIST datasets, the numbers of context pixels in testing are $\{10, 100, 300, 500, 700\}$. For CIFAR10/SVHN datasets, the numbers of context pixels in testing are $\{10, 100, 300, 500, 800, 1000\}$.

Another critical finding is reported in Fig. (6), which examines whether the learned functional priors collapse into the deterministic ones. This is based on the average computed trace of covariance matrices in learned functional priors. In MNIST dataset, SI-NPs encounter the prior collapse, and ML-NPs also obtain extremely lower trace values of covariance matrices. We attribute the prior collapse in MNIST to its most superficial structures and limited semantic diversity, which results in lower prior uncertainty. For image datasets with more complicated semantics, the computed trace values in SI-NPs retain a reasonable interval. The scale of SI-NPs' trace values coincides with the semantics complexity of image datasets: CIFAR10>SVHN>FMNIST>MNIST. Another common observation is that with context points increasing, both models' trace values of functional priors in most datasets decrease to a certain level.

The above reveals the meaning of the learned functional priors in SI-NPs: The functional prior exhibits higher uncertainty with more complicated semantics. Hence, it has the potential of generating functional samples with more diversity. For datasets with less rich semantics, the randomness of the learned functional priors plays a less critical role, and the deterministic functional representation is capable of function generation. Also, with more observations, the randomness of functional priors can be reduced.

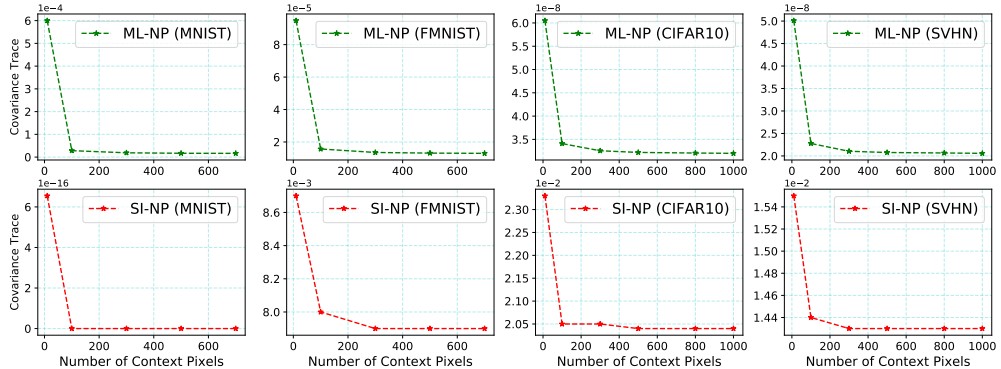

Figure 6: **Statistics of Learned Functional Priors in ML-NPs/SI-NPs.** In meta testing, we still vary the number of context points in image datasets. The trace of learned functional priors' covariance matrices $\text{Tr}[\Sigma_\vartheta(\mathcal{D}_\tau^C)]$ is computed based on $p(z|\mathcal{D}_\tau^C; \vartheta) = \mathcal{N}(z; \mu_\vartheta(\mathcal{D}_\tau^C), \Sigma_\vartheta(\mathcal{D}_\tau^C))$.

# 6 RELATED WORK

**NPs Family.** NPs are simple and flexible in model formulations, but they may suffer from underfitting. Previous research focuses more on the use of structural inductive biases in NPs. (Kim et al., 2019; 2021) improves the predictive performance by adding attention based local variables. Translation equivariance and invariance are incorporated in modeling NPs with help of convolutions (Gordon et al., 2019; Foong et al., 2020; Kawano et al., 2020). To find more compact functional representations, the contrastive loss is used to regularize (C)NPs' training (Gondal et al., 2021). Hierarchical and mixture structures are also explored to induce diverse latent variables in NPs (Wang & Van Hoof, 2020; Wang & van Hoof, 2022). Lee et al. (2020) combines boostrapping tricks with neural processes to improve expressiveness of latent variables. Besides, NPs are applied to address sequential decision-making problems (Galashov et al., 2019; Wang & Van Hoof, 2022). A summary of these models is given in Appendix (D.3).

**Expectation Maximization & Wake-Sleep Algorithms.** For log-likelihood maximization problems with incomplete observations, expectation maximization (Bishop & Nasrabadi, 2006; Balestriero et al., 2020) is a tractable approach in optimization. It consists of an E-step to optimize the distribution of latent variables and a M-step to maximize the likelihood parameters. The optimization of VAEs and variants is also built upon an EM framework (Ghojogh et al., 2021; Gao et al., 2021). Our developed algorithm can be interpreted as EM for NPs. Another family of algorithms related to our method is wake-sleep algorithms (Bornschein & Bengio, 2014; Eslami et al., 2016; Dieng & Paisley, 2019; Le et al., 2020), where self-normalized importance sampling is used in Monte Carlo estimates. In this case, our method can be also viewed as the extension of the reweighted wake-sleep (RWS) algorithm to NPs with an improvement guarantee. Another difference lies in that optimizing a learnable functional prior is of most importance in NPs, while RWS algorithms are mostly used in scenarios with a fixed prior.

# 7 CONCLUSION

**Technical Discussions.** In this paper, we study NPs family from an optimization objective perspective and analyze the inference suboptimality of vanilla NPs. Within the variational expectation maximization framework, our developed SI-NP improves the target likelihood step by step to obtain a more optimal functional prior. Besides, experimental results tell us the learned functional prior has close connections with the number of context points and complexity of function families.

**Existing Limitations.** Compared to vanilla NPs, SI-NP requires more than one Monte Carlo sample from the functional prior in training, which consumes more computations. Though intuitively, the uncertainty of the functional prior can be forward propagated to the output distribution, the influence of such uncertainty has not been mathematically studied.

**Future Extensions.** The SI-NP objective is regardless of neural architectures so that any structural inductive bias can be incorporated in modeling. The combination can theoretically produce superior functional representation baselines in the domain (Please refer to Appendix (H.5) for extensive experiments).

## ACKNOWLEDGEMENT

We thank Dharmesh Tailor for the helpful discussions. We thank Eric Nalisnick and Tim Bakker for the helpful feedback in the AMLab Seminar talks.

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

CONTENTS

# A  FREQUENTLY ASKED QUESTIONS

In this section, we list frequently asked questions from researchers who help proofread this manuscript. These raised questions are crucial, so we include more detailed discussions here. Meanwhile, we make extra efforts for readers to quickly understand our work and make slides as follows https://anonymous.4open.science/r/SI_NPs_Slides-7C94/iclr_submission_slides.pdf.

**Learnable Proposal Distributions.** The reason why we set the update of learnable proposal distribution an optional choice is that we find it difficult to balance the optimization of Eq. (12) and Eq. (31). This is non-trivial and assigning equal weights in optimization results in unstable performance in experiments (Please refer to Section (H.1) for more discussion.). So, we leave the improvement of the proposal distribution future research. Suppose a reasonable objective or at least a weight scheduling mechanism for two objectives in two E-steps can be developed to stabilize the training. In that case, SI-NPs are likely to achieve even better performance.

**Influence of Functional Prior Collapse.** Generally, the generated functions are more diverse when the functional prior does not collapse: More randomness can be reflected from the functional prior, and the uncertainty of partial observation can be propagated to the output space. However, in experiments, we observe that for functions with more superficial structures, *e.g.* MNIST dataset, the deterministic functional representation in CNPs is sufficient to generate functions of high quality. So there is no consensus on the influence of the functional prior collapse on the generation performance.

**Influence of the Number of Inference Particles in SI-NPs.** With more inference particles (latent variables), SI-NPs can avoid the prior collapse in some cases, but the cost is expensive for computations. To trade-off performance and computations, we set the required number of particles for meta training in a small quantity, *e.g.* 8 particles for image completion tasks.

**Connections between Different Optimization Objectives.** In Table (3), we summarize the connections of different objectives with the meta dataset log-likelihood $\mathcal{L}(\vartheta)$. Note that $\mathcal{L}_{\text{NP}}(\vartheta, \phi)$ is not the exact ELBO of $\mathcal{L}(\vartheta)$. $\mathcal{L}_{\text{ML-NP}}(\vartheta)$ can also be viewed as the importance weighted method with equal weights for all particles. Since the estimates of importance weights in SI-NPs exploit the target observations and consider the difference in particles, this helps improve the model's generalization capability. Though the self-normalized importance sampling makes SI-NPs' estimates over $\mathcal{L}(\vartheta)$ biased, there is a probabilistic convergence guarantee towards the true integral quantity of our interest (Please refer to **Chapter 9** in the Book "**Monte Carlo Theory, Methods and Examples**" (Owen, 2013)).

**Combination with Structural Inductive Biases.** The primary research focus is on the optimality of the different optimization objectives, and this study applies to more general stochastic processes (stationary or nonstationary cases). Theoretically, the objective of SI-NPs can be combined with most structural inductive biases in Table (4), especially when the function family is complicated. For example, the scale parameter of Matern kernels also varies with kernel values, bringing more variations in realizations. In this case, we guess a global latent variable is difficult to handle: SI-NPs and ML-NPs might encounter a similar performance bottleneck from the empirical observations in the main paper. To further improve the performance, we take the attention module (Kim et al., 2019) as an example to conduct extensive experiments in Section (H.5). Since the meta learning experiment is computationally expensive and time-consuming in training processes, we do not examine combinations with other inductive biases in this paper.

**GP Oracles in Synthetic Regression.** Note that the GP oracle is computed in the form $\ln p(y_{1:n+m}|y_{1:n}, x_{1:n+m})$, where the predictive distribution is mainly with a non-diagonal covariance matrix (suggesting $y_{n+m}$ not independent in statistics). In comparison, the log-likelihood of NPs family is computed in the form $\sum_{i=1}^{n+m} \ln p(y_i|[x_i, z]; \vartheta)$, where $y_{1:n+m}$ is conditional independent *w.r.t.* the global latent variable $z$. Such a difference in computations causes the mentioned gap. However, with a large neural model, such as the integration of attention networks in Section (H.5), the quantified performance gap of GP Oracles and the log-likelihood of NPs family is well reduced.

Table 3: A Summary of Optimization Objectives in NPs Family. We list the available optimization objectives in Section (3)/(4). For Importance Weighted Estimates, multiple Monte Carlo samples are required in meta training.

| Optimization Objective | Connection with $\mathcal{L}(\vartheta)$ in Eq. (4) | Importance Weighted Estimates |
|---|---|---|
| $\mathcal{L}_{\text{NP}}(\vartheta, \phi)$ | Approximate ELBO | ✗ |
| $\mathcal{L}_{\text{CNP}}(\vartheta)$ | Biased Estimate | ✗ |
| $\mathcal{L}_{\text{ML-NP}}(\vartheta)$ | Biased Estimate | ✓ |
| $\mathcal{L}_{\text{SI-NP}}$ (Ours) | Biased Estimate | ✓ |

## B  PROBABILISTIC GENERATIVE PROCESS IN NPs

**Definition B.1 (Exchangeable Stochastic Processes)** *We denote a probability space of functions by $(\Omega, \mathcal{F}, \mathbb{P})$. Let $\nu_{x_1,\ldots,x_N}$ be a probability measure on $\mathbb{R}^d$ with $\{x_1,\ldots,x_N\}$ a finite index set. The process is called an exchangeable stochastic process $\mathcal{S}: X \times \Omega \to \mathbb{R}^d$ such that $\nu_{x_1,\ldots,x_N}(F_1 \times \cdots \times F_N) = \mathbb{P}(\mathcal{S}_{x_1} \in F_1, \ldots, \mathcal{S}_{x_N} \in F_N)$ if it satisfies exchangeable consistency and marginalization consistency.*

Here we can translate the generative process of NPs in the following mathematical way.

$$\tau \sim p(\mathcal{T}), \quad z \sim \mathcal{N}(z; \mu_\vartheta(\mathcal{D}_\tau^C), \Sigma_\vartheta(\mathcal{D}_\tau^C))$$
$$x_i \sim p(x), \quad y_i \sim p(y|[x_i, z]; \vartheta) \quad \forall i \in \{1, 2, \ldots, n+m\} \tag{15}$$

## C  PREDICTIVE DISTRIBUTIONS IN GPs & NPs

Here we take one-dimensional deep Gaussian processes (Dai et al., 2016) as an example. With context points $\mathcal{D}^C = \{(x_i, y_i)\}_{i=1}^n$ and target points $\mathcal{D}^T = \{(x_i, y_i)\}_{i=1}^{n+m} = [x_T, y_T]$, the key to applications is the predictive distribution $p(f(x_T)|\mathcal{D}^C, x_T) = \mathcal{N}(y_T; \mu_T, \Sigma_T)$. The conditional mean $\mu_T$ and covariance $\Sigma_T$ functions in Eq. (16) are permutation invariant to the order of context points.

$$\mu_T = m_\theta(x_T) + \Sigma_{T,C}\Sigma_{C,C}^{-1}(y_C - m_\theta(x_C))$$
$$\Sigma_T = \Sigma_{T,T} - \Sigma_{T,C}\Sigma_{C,C}^{-1}\Sigma_{C,T} \tag{16}$$

Here the covariance matrix denoted by $\Sigma$ is computed with the context input $x_C = (x_1, \ldots, x_n) \in \mathbb{R}^{n \times d}$, the target input $x_T = (x_1, \ldots, x_{n+m}) \in \mathbb{R}^{(n+m) \times d}$, and a kernel function $\psi$, *e.g.* $[\Sigma_{C,C}]_{i,j} = \psi(x_i, x_j)$, $m_\theta$ is the mean function $m_\theta$, and the context output is $y_C = (y_1, \ldots, y_n) \in \mathbb{R}^n$. Nevertheless, the computation of matrix inversion in Eq. (16) makes the runtime complexity as expensive as $\mathcal{O}((n+m)^3)$.

## D  NPs FORMULATION & STRUCTURAL INDUCTIVE BIASES

### D.1  PRIOR, POSTERIOR & PROPOSAL DISTRIBUTIONS

Since fast adaptation is achieved in an amortized way, which reduces the gradient updates *w.r.t.* model parameters to learning function specific latent variables with meta trained neural networks.

**Definition D.1 (Permutation Invariant Functions)** *Let $\mathcal{S}_n$ be an $n$-element permutation group. For any permutation operator $g \in \mathcal{S}_n$, the function is a bijective mapping from the order set $\{1, 2, \ldots, n\}$ to itself:*

$$g: [1, 2, \ldots, n] \to [g_1, g_2, \ldots, g_n].$$

*Then given a set of data points $\{x_1, x_2, \ldots, x_n\}$, the function $\Phi$ is said to be a permutation invariant function if the following equation is satisfied $\forall g \in \mathcal{S}_n$*

$$\Phi(g \circ [x_1, x_2, \ldots, x_n]) = \Phi([x_{g_1}, x_{g_2}, \ldots, x_{g_n}]) = \Phi([x_1, x_2, \ldots, x_n]).$$

The context points and the target points are treated as sets, so the amortized network should be permutation invariant *w.r.t.* the order of data points.

**Approximate Posterior Distribution.** This is denoted by $q_\phi(z|\mathcal{D}_\tau^T)$ in this paper. Usually, the approximate posterior is used in NPs (Garnelo et al., 2018b; Kim et al., 2019) and works as a proxy for the non-analytical exact posterior $p(z|\mathcal{D}_\tau^T; \vartheta)$.

**Prior Distribution.** This is denoted by $p(z|\mathcal{D}_\tau^C; \vartheta)$ in this paper. Unlike the approximate prior $q_\phi(z|\mathcal{D}_\tau^C)$ used in NPs, we use an exact functional prior in SI-NPs.

**Proposal Distribution.** This is denoted by $q_\eta(z|\mathcal{D}_\tau^T)$ in this paper. The role of the proposal distribution resembles that of the approximate posterior in NPs. It is used to sample latent variables and enables the computation of the importance weights in NPs.

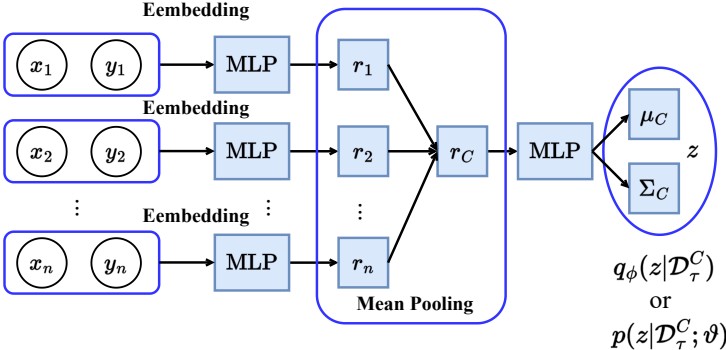

**(a) Approximate or Learned Functional Priors**

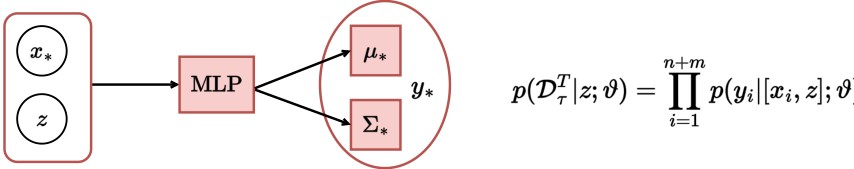

**(b) Generative Distributions**

Figure 7: Computational Diagram of Vanilla NPs. The blue one and the pink one are, respectively, the encoder and the decoder structures in vanilla NPs. The approximate posterior, the approximate prior, the learned prior, and the proposal distributions are in the same neural structure as the encoder. For the generative distribution in the form of the decoder, the diagram shows one instance $(x_*, y_*) \in \mathcal{D}_\tau^T$ in a generation and the sampled global latent variable $z \sim p(z|\mathcal{D}_\tau^C; \vartheta)$ or $z \sim q_\phi(z|\mathcal{D}_\tau^C)$ is shared across all data points in one realization.

In vanilla NPs, the latent variable $z$ is inferred from a set of data points, and the distributions $q_\phi(z)$ and $q_\phi(z|\mathcal{D}_\tau^C)$ are learned via functions permutation invariant to the order of data points. An example module to parameterize $q_\phi(z|\mathcal{D}_\tau^C) = \mathcal{N}(z; \mu_C, \Sigma_C)$ can be Eq. (17) with $\bigoplus$ a mean pooling operator and $\{h_\phi, g_\phi\}$ encoder networks.

$$r_i = h_\phi([x_i, y_i]) \; \forall (x_i, y_i) \in \mathcal{D}_\tau^C, \quad r_C = \bigoplus_{i=1}^{N} r_i, \quad [\mu_C, \Sigma_C] = g_\phi(r_C) \tag{17}$$

The same with that in traditional stochastic processes, a realisation corresponds to a sampled function $f(X)$ generated in a sequential way: $z \sim q_\phi(z|\mathcal{D}_\tau^C), f(X) \sim p(Y|X, z; \vartheta)$.

*Since this paper aims to study the influence of different optimization objectives concerning the NP models, we retain the basic model set-up for all baselines.* The Fig. (7) exhibits the neural architecture of the vanilla NPs, and such a structure is also shared with ML-NPs and SI-NPs in our paper.

As previously mentioned, these methods can be straightforwardly incorporated with other structural inductive biases (Kim et al., 2019; Gordon et al., 2019; Gondal et al., 2021; Wang & van Hoof, 2022). For example, in the following section, we augment all methods with attention networks (Kim et al., 2019) and examine the resulting performance.

## D.2 Approximate ELBOs in NPs & Proof of Remark 1

$$\ln p(\mathcal{D}_\tau^T | \mathcal{D}_\tau^C; \vartheta) = \ln \int p(\mathcal{D}_\tau^T | z; \vartheta) p(z | \mathcal{D}_\tau^C; \vartheta) dz \quad (18a)$$

$$\geq \mathbb{E}_{q_\phi(z)} \left[ \ln p(\mathcal{D}_\tau^T | z; \vartheta) \right] - D_{KL} \left[ \underbrace{q_\phi(z)}_{\text{Approximate Posterior}} \| \underbrace{p(z | \mathcal{D}_\tau^C; \vartheta)}_{\text{Functional Prior}} \right] = \mathcal{L}_{\text{ELBO}}(\vartheta, \phi) \quad (18b)$$

$$\approx \mathbb{E}_{q_\phi(z)} \left[ \ln p(\mathcal{D}_\tau^T | z; \vartheta) \right] - D_{KL} \left[ \underbrace{q_\phi(z)}_{\text{Approximate Posterior}} \| \underbrace{q_\phi(z | \mathcal{D}_\tau^C)}_{\text{Approximate Prior}} \right] = \mathcal{L}_{\text{NP}}(\vartheta, \phi) \quad \square \quad (18c)$$

For Eq. (18.b), remember that *w.r.t.* these VAE-like methods, it is hard to guarantee the improvement of the evidence in each iteration when optimizing ELBO due to the existence of a posterior approximation gap. Meanwhile, the form of the functional prior is unknown.

**Proof D.1 (Remark 1)** *Vanilla NPs directly replace the real functional prior by the approximate one $q_\phi(z | \mathcal{D}_\tau^C)$ and introduce the consistent regularizer in Eq. (7). We further introduce the prior approximation gap in Eq. (19), in which the sign is undetermined.*

$$\mathcal{L}_{NP}(\vartheta, \phi) = \mathcal{L}_{ELBO}(\vartheta, \phi) + \underbrace{\mathbb{E}_{q_\phi(z)} \left[ \ln \frac{q_\phi(z | \mathcal{D}_\tau^C)}{p(z | \mathcal{D}_\tau^C; \vartheta)} \right]}_{\text{Prior Approximation Gap}} \quad (19)$$

*Based on decomposition in Eq. (6)/(19), we can find that there exists no consistent monotonic relationship between $\mathcal{L}(\vartheta)$ and $\mathcal{L}_{NP}(\vartheta, \phi)$. The invalid ELBO makes the optimization w.r.t. $\mathcal{L}_{NP}(\vartheta, \phi)$ not always improve log-likelihood $\mathcal{L}(\vartheta)$.*

$$\mathcal{L}(\vartheta) \geq \mathcal{L}_{ELBO}(\vartheta, \phi), \quad \mathcal{L}(\vartheta) \ngeq \mathcal{L}_{NP}(\vartheta, \phi) \quad \forall \vartheta \in \Theta \text{ and } \phi \in \Phi \quad (20)$$

*On the left side of Eq. (20), the ELBO of NPs can be bounded by the log-likelihood of meta learning dataset. Optimizing the ELBO can guarantee the optimal or local optimal solution when the approximation gap and the amortization gap are well closed. However, the right side of Eq. (20) implies the previous commonly-used strategies, e.g. normalizing flows for richer variational posterior distributions (Rezende & Mohamed, 2015), auxiliary variables for augmented variational posterior distributions (Maaløe et al., 2016) and more flexible prior distributions (Tomczak & Welling, 2018), to close VAEs inference gaps (Cremer et al., 2018) will not guarantee the performance improvement in a theoretical sense. In other words, the consistent regularizer in NPs is ill-posed for optimization.*

## D.3 Summary of NPs Family

In this part, we summarize the encoder and decoders for typical NP variants and point out the inductive biases behind them. The information is summarized in Table (4). We can see a variety of inductive biases can be used to improve NPs, most of them from the model structure perspective rather than the optimization objective.

Table 4: Summary of Typical Neural Process Related Models (Meta-Testing Scenarios). The recognition model and the generative model respectively correspond to the encoder and the decoder in the family of neural processes. Here $[x_C, y_C]$ are context data points and we consider $(x_*, y_*)$ a data point in target dataset.

| Models | Recognition Model | Generative Model | Inductive Bias |
|---|---|---|---|
| CNP (Garnelo et al., 2018a) | $z = f_\phi(x_C, y_C)$ | $p_\theta(y_* \vert [x_*, z])$ | conditional functional |
| NP (Garnelo et al., 2018b) | $q_\phi(z \vert [x_C, y_C])$ | $p_\theta(y_* \vert [x_*, z])$ | global functional |
| ANP (Kim et al., 2019; 2021) | $q_{\phi_1}(z \vert [x_C, y_C])$ $f_{\phi_2}(z_* \vert [x_C, y_C], x_*)$ | $p_\theta(y_* \vert [x_*, z, z_*])$ | global functional local embedding |
| FCRL (Gondal et al., 2021) | $f_\phi(z \vert [x_C, y_C])$ | $p_\theta(y_* \vert [x_*, z])$ | contrastive functional |
| ConvNP (Foong et al., 2020) | $p_\phi(z \vert [x_C, y_C])$ | $p_\theta(y_* \vert [x_*, z])$ | convolutional functional |
| Conv-CNP (Gordon et al., 2019) | $f_\phi(z_* \vert [x_C, y_C], x_*)$ | $p_\theta(y_* \vert [x_*, z_*])$ | convolutional functional |
| FNP (Louizos et al., 2019) | $f_\phi(z_* \vert [x_C, y_C], x_*)$ | $p_\theta(y_* \vert z_*)$ | latent DAG |

### D.4 Inference Gaps

In this section, we apply the trick of inference gap decomposition (Cremer et al., 2018) to understand vanilla NPs. Here we denote the approximate inference gap by $D_{KL}^{\text{AI}}$ and the posterior approximation gap by $D_{KL}^{\text{PA}}$ in Table (5).

Since it is infeasible to obtain the exact form for the functional posterior, we cannot directly close the mentioned approximate gap, and approximate methods are used to learn them. In vanilla NPs, the consistent regularizer in Eq. (7) works as the surrogate for the ELBO. But based on our analysis, closing this surrogate gap cannot lead to a theoretically optimal solution of functional priors.

Table 5: Inference Gaps in vanilla NPs. The $\downarrow$ indicates the minimization to obtain the optimal inference solution. The sign $--$ means not applicable in deriving equivalent KL divergence form. The sign $*$ indicates the optimal posterior approximation in the family of variational distributions $\phi^* = \arg\min_{\phi \in \Phi} D_{KL}\left[q_\phi(z) \parallel p(z \vert \mathcal{D}_\tau^T; \vartheta^*)\right]$. Here $\vartheta^*$ consists of the optimal parameters in priors $p(z \vert \mathcal{D}_\tau^C; \vartheta^*)$ and the conditional distribution $p(\mathcal{D}_\tau^T \vert z; \vartheta^*)$.

| Terms | Optimization Objective | KL Divergence or Gaps |
|---|---|---|
| Approximate Inference Posterior Approximation Amortization | $\mathcal{L}(\vartheta^*) - \mathcal{L}_{\text{ELBO}}(\vartheta^*, \phi) \downarrow$ $\mathcal{L}(\vartheta^*) - \mathcal{L}_{\text{ELBO}}(\vartheta^*, \phi^*) \downarrow$ $\mathcal{L}_{\text{ELBO}}(\vartheta^*, \phi^*) - \mathcal{L}_{\text{ELBO}}(\vartheta^*, \phi) \downarrow$ | $D_{KL}\left[q_\phi(z) \parallel p(z^* \vert \mathcal{D}_\tau^T; \vartheta)\right]$ $D_{KL}\left[q_{\phi^*}(z) \parallel p(z \vert \mathcal{D}_\tau^T; \vartheta^*)\right]$ $D_{KL}^{\text{AI}} - D_{KL}^{\text{PA}}$ |
| NP Prior Approximation | $\vert \mathcal{L}_{\text{ELBO}}(\vartheta^*, \phi^*) - \mathcal{L}_{\text{NP}}(\vartheta^*, \phi^*) \vert$ | $-$ |
| Surrogate Likelihood | $\mathcal{L}(\vartheta^*) - \mathcal{L}(\vartheta; \vartheta_k) \downarrow$ | $\mathcal{L}(\vartheta^*) - \mathcal{L}(\vartheta_H)$ |

## E Formulation of Variational Expectation Maximization Method

In this section, we detail the progress of optimizing the NP model with the help of variational expectation maximization algorithms. The concept of meta learning surrogate functions is introduced, and NPs are verified. Meanwhile, the improvement guarantee as well as other concerning technical points are included to better understand our method.

### E.1 Proof of Improvement Guarantee using Variational Expectation Maximization

#### E.1.1 Meta Learning Surrogate Function

**Definition E.1 (Surrogate Function)** [4] *Given the objective function $f(\vartheta)$ to maximize, surrogate functions $g(\vartheta; \vartheta_k)$ w.r.t. $f(\vartheta)$ are a family of functions with the following properties.*

- *Both $f(\vartheta)$ and $g(\vartheta; \vartheta_k)$ are $C^2$-functions, which means $f(\vartheta)$ has first order and second order derivatives of those functions exist and these are smooth in the domain of the function.*

- *The following two formulas hold:*

$$g(\vartheta; \vartheta_k) \leq f(\vartheta) \,\forall \vartheta; \quad g(\vartheta_k; \vartheta_k) = f(\vartheta_k). \tag{21}$$

When the objective function $f(\vartheta)$ to maximize is complicated, *e.g.* multi-modal likelihood functions, a surrogate function $g(\vartheta; \vartheta_k)$ enables an easier proxy implementation with convergence guarantee to at least the local optimal. To see this point, recall the properties of the surrogate function $g(\vartheta_k; \vartheta_k) = f(\vartheta_k)$. The update rule for the surrogate function follows that $\vartheta_{k+1} = \arg\max_\vartheta g(\vartheta; \vartheta_k)$. And this results in $f(\vartheta_{k+1}) \geq g(\vartheta_{k+1}; \vartheta_k) \geq f(\vartheta_k)$.

Recall the following function $\mathcal{L}(\vartheta; \vartheta_k)$ in the main paper.

$$\mathcal{L}(\vartheta; \vartheta_k) = \sum_{\tau \in \mathcal{T}} \mathbb{E}_{p(z|\mathcal{D}_\tau^T; \vartheta_k)} \left[ \ln p(\mathcal{D}_\tau^T, z | \mathcal{D}_\tau^C; \vartheta) - \ln p(z | \mathcal{D}_\tau^T; \vartheta_k) \right] \tag{22}$$

The expectation operation in the $k$-th iteration step corresponds to `E-step` : $q_\phi(z|\mathcal{D}_\tau^C) = p_{\vartheta_k}(z|\mathcal{D}_\tau^C)$, while the maximization operation in the $k$-th iteration step updates the parameter as `M-step` : $\vartheta_{k+1} = \arg\max_\vartheta \mathcal{L}(\vartheta; \vartheta_k)$.

$$\underbrace{\ln p(\mathcal{D}_\tau^T | \mathcal{D}_\tau^C; \vartheta_k)}_{\text{Model Evidence}} \underset{\text{E-step}}{=} \mathcal{L}(\vartheta_k; \vartheta_k) \underset{\text{M-step}}{\leq} \mathcal{L}(\vartheta_{k+1}; \vartheta_k) \tag{23a}$$

$$\leq \mathcal{L}(\vartheta_{k+1}; \vartheta_k) + D_{KL}[p(z|\mathcal{D}_\tau^T; \vartheta_k) \,\|\, p(z|\mathcal{D}_\tau^T; \vartheta_{k+1})] = \underbrace{\ln p(\mathcal{D}_\tau^T | \mathcal{D}_\tau^C; \vartheta_{k+1})}_{\text{Model Evidence}} \quad \square \tag{23b}$$

#### E.1.2 Improvement Guarantee

As illustrated in Eq.s (23), the surrogate function $\mathcal{L}(\vartheta; \vartheta_k)$ is bounded by two log-likelihoods. Over the process of iterations, the log-likelihood is gradually increased to the final convergence.

$$\ln p(\mathcal{D}_\tau^T | \mathcal{D}_\tau^C; \vartheta_1) \leq \mathcal{L}(\vartheta_2; \vartheta_1) \leq \ln p(\mathcal{D}_\tau^T | \mathcal{D}_\tau^C; \vartheta_2) \leq \cdots \leq \mathcal{L}(\vartheta_H; \vartheta_{H-1}) \leq \ln p(\mathcal{D}_\tau^T | \mathcal{D}_\tau^C; \vartheta_H) \tag{24}$$

This indicates that the finally updated parameters of the surrogate function are exactly (sub-)optimal ones for the evidence. Based on these rules, directly optimizing the surrogate function step by step can theoretically guarantee the finding of optimal or at least a local optimal parameters.

### E.2 Importance Sampling in a Variational EM Algorithm

Though the conditional marginal distribution $p(\mathcal{D}_\tau^T | \mathcal{D}_\tau^C; \vartheta_k)$ is not analytical, the importance sampling trick can be used to estimate the result with the help of a proposal distribution $q_\eta(z|\mathcal{D}_\tau^T)$.

---

[4]Note that the surrogate function above is defined in the Minorize-Maximization (MM) algorithm sense (Hunter & Lange, 2004).

$$p(\mathcal{D}_\tau^T|\mathcal{D}_\tau^C;\vartheta_k) = \int p(\mathcal{D}_\tau^T, z|\mathcal{D}_\tau^C;\vartheta_k)dz = \int q_\eta(z|\mathcal{D}_\tau^T)\frac{p(\mathcal{D}_\tau^T, z|\mathcal{D}_\tau^C;\vartheta_k)}{q_\eta(z|\mathcal{D}_\tau^T)}dz$$

$$\approx \frac{1}{B}\sum_{b=1}^{B}\frac{p(\mathcal{D}_\tau^T, z^{(b)}|\mathcal{D}_\tau^C;\vartheta_k)}{q_\eta(z^{(b)}|\mathcal{D}_\tau^T)} = \frac{1}{B}\sum_{b=1}^{B}\omega^{(b)}, \qquad (25)$$

$$\text{with } z^{(b)} \sim q_\eta(z|\mathcal{D}_\tau^T) \text{ and } \omega^{(b)} = \frac{p(\mathcal{D}_\tau^T, z^{(b)}|\mathcal{D}_\tau^C;\vartheta_k)}{q_\eta(z^{(b)}|\mathcal{D}_\tau^T)}$$

Especially, the joint distribution is computed via the decomposition that $p(\mathcal{D}_\tau^T, z^{(b)}|\mathcal{D}_\tau^C;\vartheta_k) = p(z^{(b)}|\mathcal{D}_\tau^C;\vartheta_k)p(\mathcal{D}_\tau^T|z^{(b)};\vartheta_k)$ with $p(\mathcal{D}_\tau^T|z^{(b)};\vartheta_k) = \prod_{i=1}^{n+m} p(y_i|[x_i, z^{(b)}];\vartheta_k)$.

With the above equation, the intractable optimization objective is transformed into a feasible one.

$$\mathcal{L}_{\text{EM}}(\vartheta;\vartheta_k) = \mathbb{E}_{p(z|\mathcal{D}_\tau^T;\vartheta_k)}\ln p(\mathcal{D}_\tau^T, z|\mathcal{D}_\tau^C;\vartheta) = \int \frac{p(\mathcal{D}_\tau^T, z|\mathcal{D}_\tau^C;\vartheta_k)}{p(\mathcal{D}_\tau^T|\mathcal{D}_\tau^C;\vartheta_k)}\ln p(\mathcal{D}_\tau^T, z|\mathcal{D}_\tau^C;\vartheta)dz$$

$$(26a)$$

$$= \int q_\eta(z|\mathcal{D}_\tau^T)\frac{p(\mathcal{D}_\tau^T, z|\mathcal{D}_\tau^C;\vartheta_k)}{q_\eta(z|\mathcal{D}_\tau^T)p(\mathcal{D}_\tau^T|\mathcal{D}_\tau^C;\vartheta_k)}\ln p(\mathcal{D}_\tau^T, z|\mathcal{D}_\tau^C;\vartheta)dz$$

$$(26b)$$

$$\approx \frac{1}{B}\sum_{b=1}^{B}\frac{\omega^{(b)}}{p(\mathcal{D}_\tau^T|\mathcal{D}_\tau^C;\vartheta_k)}\ln p(\mathcal{D}_\tau^T, z^{(b)}|\mathcal{D}_\tau^C;\vartheta)$$

$$(26c)$$

$$= \sum_{b=1}^{B}\frac{\omega^{(b)}}{\sum_{b'=1}^{B}\omega^{(b')}}\ln p(\mathcal{D}_\tau^T, z^{(b)}|\mathcal{D}_\tau^C;\vartheta) = \sum_{b=1}^{B}\hat{\omega}^{(b)}\ln p(\mathcal{D}_\tau^T, z^{(b)}|\mathcal{D}_\tau^C;\vartheta) = \mathcal{L}_{\text{SI-NP}}(\vartheta, \eta;\vartheta_k)$$

$$(26d)$$

We can expand the term inside the expectation in Eq (26.a) as follows.

$$p(\mathcal{D}_\tau^T, z^{(b)}|\mathcal{D}_\tau^C;\vartheta) = p(z^{(b)}|\mathcal{D}_\tau^C;\vartheta)p(\mathcal{D}_\tau^T|z^{(b)};\vartheta) \qquad (27)$$

As for the exact posterior $p(z|\mathcal{D}_\tau^T;\vartheta_k)$, we can get the following expansion.

$$p(z|\mathcal{D}_\tau^T;\vartheta_k) = \frac{p(z, \mathcal{D}_\tau^T;\vartheta_k)}{\int p(z, \mathcal{D}_\tau^T;\vartheta_k)dz} = \frac{p(z|\mathcal{D}_\tau^C;\vartheta_k)p(\mathcal{D}_\tau^T|z;\vartheta_k)}{\int p(z|\mathcal{D}_\tau^C;\vartheta_k)p(\mathcal{D}_\tau^T|z;\vartheta_k)dz} \qquad (28)$$

The distribution has the complicated denominator and this makes it infeasible to directly sample from the conditional distribution.

### E.3 OPTIMIZATION OBJECTIVE WITH PROPOSAL DISTRIBUTIONS

This subsection shows the mentioned optional optimization step E-step #2 in Algorithm (1).

### E.3.1 UPDATE OF PROPOSAL DISTRIBUTIONS (OPTIONAL)

The use of proposal distribution $q_\eta$ enables sampling $z$ for Monte Carlo estimates. Another role of the proposal distribution is to work as a proxy for the posterior $p(z|\mathcal{D}_\tau^T;\vartheta_k)$, and the variance

$\mathbb{V}_{q_\eta} \left[ \frac{p(z|\mathcal{D}_\tau^T;\vartheta_k)}{q_\eta(z|\mathcal{D}_\tau^T)} \ln p(\mathcal{D}_\tau^T, z|\mathcal{D}_\tau^C; \vartheta) \right]$ is expected to be lower for stable training. So a reasonable optimization objective is to get two distributions as close as each other, *e.g.* minimizing the Kullback–Leibler divergence $D_{KL}[p(z|\mathcal{D}_\tau^T;\vartheta_k) \parallel q_\eta(z|\mathcal{D}_\tau^T)]$.

This treatment is equivalent to wake phase updates in (Bornschein & Bengio, 2014). We can obtain the optimization objective on the right side of Eq. (29) with the help of the self-normalized importance sampling.

$$\min_\eta D_{KL}[p(z|\mathcal{D}_\tau^T;\vartheta_k) \parallel q_\eta(z|\mathcal{D}_\tau^T)]$$

$$\Leftrightarrow \min_\eta \mathcal{L}_{\text{KL}}(\eta; \eta_{k-1}, \vartheta_k) = -\sum_{b=1}^{B} \hat{\omega}^{(b)} \ln q_\eta(z^{(b)}|\mathcal{D}_\tau^T) \qquad (29)$$

Here the self-normalized importance weights $\{\hat{\omega}^{(b)}\}_{b=1}^B$ inside the above equation are the same as that in Eq. (12). Also note that the the denominator $\frac{1}{B} \sum_{b'=1}^{B} \omega^{(b')}$ of the weight relates to the estimate of $p(\mathcal{D}_\tau^T|\mathcal{D}_\tau^C)$, which has biases with limited samples at the beginning of training. However, with the improvement of approximation, the bias can be decreased accordingly (Zimmermann et al., 2021).

### E.3.2 DERIVATION OF THE PROPOSAL UPDATE OBJECTIVE

It is trivial to see the following equation since the term $\mathbb{E}_{p(z|\mathcal{D}_\tau^T;\vartheta_k)} \left[ p(z|\mathcal{D}_\tau^T;\vartheta_k) \right]$ is a constant.

$$\min_\eta D_{KL}[p(z|\mathcal{D}_\tau^T;\vartheta_k) \parallel q_\eta(z|\mathcal{D}_\tau^T)] \Leftrightarrow \min_\eta -\mathbb{E}_{p(z|\mathcal{D}_\tau^T;\vartheta_k)}[\ln q_\eta(z|\mathcal{D}_\tau^T)] \qquad (30)$$

Once again, we apply self-normalized importance sampling to the right side of Eq. (30). With the same set of sampled latent variables, the reweighted objective *w.r.t.* the proposal distribution can be derived.

$$\min_\eta -\mathbb{E}_{p(z|\mathcal{D}_\tau^T;\vartheta_k)}[\ln q_\eta(z|\mathcal{D}_\tau^T)] \approx -\sum_{b=1}^{B} \hat{\omega}^{(b)} \ln q_\eta(z^{(b)}|\mathcal{D}_\tau^T) = \mathcal{L}_{\text{KL}}(\eta; \eta_{k-1}, \vartheta_k) \qquad (31)$$

### E.4 GRADIENT ESTIMATES IN VARIATIONAL EM

In the `E-step #2`, note that the model parameter is fixed as $\vartheta_k$, we can estimate the gradient of $\eta$ *w.r.t.* $\mathcal{L}_{\text{KL}}(\eta; \eta_{k-1}, \vartheta_k)$ in the following way. This operation is to close the divergence between the proposal distribution and the exact posterior distribution.

$$\frac{\partial \mathcal{L}_{\text{KL}}(\eta; \eta_{k-1}, \vartheta_k)}{\partial \eta} = \sum_{b=1}^{B} \hat{\omega}^{(b)} \left( \frac{\partial \ln q_\eta(z^{(b)}|\mathcal{D}_\tau^T)}{\partial \eta} \right) \qquad (32)$$

In the `M-step`, note that the normalized importance weights are constant, the proposal distribution is fixed, and the gradient *w.r.t.* $\mathcal{L}_{\text{SI-NP}}(\vartheta; \eta_k, \vartheta_k)$ can be estimated in a straightforward way as follows.

$$\frac{\partial \mathcal{L}_{\text{SI-NP}}(\vartheta; \eta_k, \vartheta_k)}{\partial \vartheta} = \sum_{b=1}^{B} \hat{\omega}^{(b)} \left( \frac{\partial \ln p(z^{(b)}|\mathcal{D}_\tau^C; \vartheta)}{\partial \vartheta} + \frac{\partial \ln p(\mathcal{D}_\tau^T|z^{(b)}; \vartheta)}{\partial \vartheta} \right) \qquad (33)$$

We can see the role of normalized importance weights in gradient estimates from Eq. (33): remember that $\hat{\omega}^{(b)} \propto p(\mathcal{D}_\tau^T|z^{(b)}; \vartheta_k)$ in the $k$-th iteration when the functional prior works as the proposal distribution, so the more gradients will be allocated to those particles with higher generative likelihoods. Meanwhile, the normalized importance weights influence the optimization of the functional prior distribution so that the functional prior can match the best set of particles well.

# F CONNECTION WITH CNPs

## F.1 PRIOR COLLAPSE IN SI-NPs WITH ONE MONTE CARLO SAMPLE

**Theorem 1 (L'Hôpital's Rule (Hospital, 1696))** *Let $f(x)$ and $g(x)$ be two functions differentiable on an open interval $\mathcal{I}$ except possibly at a point $c$ contained in $\mathcal{I}$. If $\lim_{x \to c} f(x) = \lim_{x \to c} g(x) = \infty$, and $\left(\frac{1}{g(x)f(x)}\right)' \neq 0$ with $\forall x \in \mathcal{I}$ and $x \neq c$, we can have the following limit equation.*

$$\lim_{x \to c} f(x) - g(x) = \lim_{x \to c} \frac{\frac{1}{g(x)} - \frac{1}{f(x)}}{\frac{1}{g(x)f(x)}} = \lim_{x \to c} \frac{\left(\frac{1}{g(x)} - \frac{1}{f(x)}\right)'}{\left(\frac{1}{g(x)f(x)}\right)'} \tag{34}$$

**Proof F.1 (Proposition 3)** *Note that in our default setup of SI-NPs, the functional prior works as the proposal distribution to sample the latent variable.*

*Let $z \in \mathbb{R}^d$ be the latent variable for a diagonal Gaussian conditional prior $p(z|\mathcal{D}_\tau^C; \vartheta) = \mathcal{N}(z; \mu_\vartheta(\mathcal{D}_\tau^C), \Sigma_\vartheta(\mathcal{D}_\tau^C))$. Here the learned mean and the covariance matrix are simply denoted by $\mu_\vartheta = [\mu_1, \ldots, \mu_d]^T \in \mathbb{R}^d$ and $\Sigma_\vartheta = diag\left[\sigma_1^2, \ldots, \sigma_d^2\right]$.*

*With one Monte Carlo sample $\hat{z} = [\hat{z}_1, \ldots, \hat{z}_d]^T$ from the conditional prior, it can be written as $\hat{z} = \mu_\vartheta + \hat{\epsilon}\Sigma_\vartheta^{\frac{1}{2}}, \hat{\epsilon} \sim \mathcal{N}(0, \mathcal{I}_d)$ with help of a reparameterization trick (Kingma & Welling, 2013).*

$$\mathbb{E}_{p(z|\mathcal{D}_\tau^C; \vartheta_k)}\left[\ln p(z|\mathcal{D}_\tau^C; \vartheta)\right] \approx \ln p(z|\mathcal{D}_\tau^C; \vartheta) = -\frac{1}{2}\ln(2\pi) + \sum_{i=1}^{d}\left[-\ln\sigma_i - \frac{(\mu_i - \hat{z}_i)^2}{2\sigma_i^2}\right] \tag{35}$$

*Eq. (35) is the result of one Monte Carlo estimate, termed as the collapse term in the main paper. Built up on these, we rewrite the SI-NP optimization objective to maximize as Eq. (36).*

$$\mathcal{L}_{\textit{SI-NP}} = \mathbb{E}_{p(z|\mathcal{D}_\tau^C; \vartheta)}\left[\ln p(\mathcal{D}_\tau^T|z; \vartheta)\right] + \mathbb{E}_{p(z|\mathcal{D}_\tau^C; \vartheta_k)}\left[\ln p(z|\mathcal{D}_\tau^C; \vartheta)\right]$$

$$\approx \sum_{i=1}^{n+m} \ln p(y_i|[x_i, \mu_\vartheta + \hat{\epsilon}\Sigma_\vartheta^{\frac{1}{2}}; \vartheta]) - \left(\frac{1}{2}\ln(2\pi) + \sum_{i=1}^{d}\left[\ln\sigma_i + \frac{(\mu_i - \hat{z}_i)^2}{2\sigma_i^2}\right]\right) \tag{36}$$

*Now we prove that when the learned variance parameter $\{\sigma_i\}_{i=1}^d$ collapse into the value zero, the optimization objective in Eq. (35) and Eq. (36) can be maximized. To simplify the notation, we put the mean variable $\mu_\vartheta$ aside, focus more on the variance variable $\Sigma_\vartheta$ and let the value $\frac{\mu_i - \hat{z}_i}{2}$ denoted by $\kappa_i$. We can directly apply **L'Hôpital's Rule** in **Theorem (1)** to these quantities and obtain the following equations.*

$$\lim_{\sigma_i \to 0} -\ln\sigma_i - \frac{\kappa_i}{\sigma_i^2} = \lim_{\sigma_i \to 0} \frac{\frac{\sigma_i^2}{\kappa_i} + \frac{1}{\ln\sigma_i}}{-\frac{\sigma_i^2}{\kappa_i \ln\sigma_i}} = \lim_{\sigma_i \to 0} \frac{\sigma_i^2 \ln\sigma_i + \kappa_i}{-\sigma_i^2}$$

$$= \lim_{\sigma_i \to 0} \frac{\left(\sigma_i^2 \ln\sigma_i + \kappa_i\right)'}{\left(-\sigma_i^2\right)'} = \lim_{\sigma_i \to 0} \frac{2\sigma_i \ln\sigma_i + \sigma_i}{-2\sigma_i} = \lim_{\sigma_i \to 0} -\ln\sigma_i - \frac{1}{2} = +\infty \tag{37}$$

*Putting them together, the Gaussian latent variable will finally collapse into a Dirac delta distribution and this demonstrates the equivalence between SI-NP with one Monte Carlo sample and CNP.*

## F.2 EFFECT OF ADJUSTING COEFFICIENTS OF THE PRIOR REGULARIZER

**Remark 2** *Either increasing the number of Monte Carlo samples or lower the weight of the collapse term in SI-NPs can effectively avoid the prior collapse.*

With increase of Monte Carlo samples, we can see the scale of the generative term outweights that of the collapse term, which indicates more weights are put in the gradient *w.r.t.* the variance parameters to maximize the generative log-likelihood. In this way, it can naturally avoid the prior collapse caused by the second term.

Next, we analyze the influence of the coefficients $\alpha$ with $\alpha \in (0,1)$ for the generative and functional prior likelihoods. To this end, we go back to the original meta learning objective and introduce the $\alpha$-dependent one as $\mathcal{L}(\vartheta; \alpha)$ in Eq. (38).

$$\max_{\vartheta} \mathbb{E}_{p(z|\mathcal{D}_\tau^T; \vartheta_k)} \left[ \ln p(\mathcal{D}_\tau^T|z; \vartheta) \right] + (1-\alpha)\mathbb{E}_{p(z|\mathcal{D}_\tau^T; \vartheta_k)} \left[ \ln p(z|\mathcal{D}_\tau^C; \vartheta) \right] = \mathcal{L}(\vartheta; \alpha)$$

$$\Leftrightarrow \max_{\vartheta} \mathbb{E}_{p(z|\mathcal{D}_\tau^T; \vartheta_k)} \left[ \ln p(\mathcal{D}_\tau^T|z; \vartheta) \right] + (1-\alpha)\mathbb{E}_{p(z|\mathcal{D}_\tau^T; \vartheta_k)} \left[ \ln \frac{p(z|\mathcal{D}_\tau^C; \vartheta)}{p(z|\mathcal{D}_\tau^T; \vartheta_k)} + \ln p(z|\mathcal{D}_\tau^T; \vartheta_k) \right]$$

$$\Leftrightarrow \max_{\vartheta} \mathbb{E}_{p(z|\mathcal{D}_\tau^T; \vartheta_k)} \left[ \ln p(\mathcal{D}_\tau^T|z; \vartheta) \right] + (1-\alpha)D_{KL} \left[ p(z|\mathcal{D}_\tau^T; \vartheta_k) \parallel p(z|\mathcal{D}_\tau^C; \vartheta) \right]$$

$$+ (1-\alpha)\underbrace{\mathbb{E}_{p(z|\mathcal{D}_\tau^T; \vartheta_k)} \left[ p(z|\mathcal{D}_\tau^T; \vartheta_k) \right]}_{\text{Constant}}$$

$$\Leftrightarrow \max_{\vartheta} \mathbb{E}_{p(z|\mathcal{D}_\tau^T; \vartheta_k)} \left[ \ln p(\mathcal{D}_\tau^T|z; \vartheta) \right] + (1-\alpha)D_{KL} \left[ p(z|\mathcal{D}_\tau^T; \vartheta_k) \parallel p(z|\mathcal{D}_\tau^C; \vartheta) \right]$$

$$\tag{38}$$

As noticed in $\mathcal{L}(\vartheta; \alpha)$, the KL divergence constraint enforces the functional prior $p(z|\mathcal{D}_\tau^C; \vartheta)$ to be close to the learned functional posterior $p(z|\mathcal{D}_\tau^T; \vartheta_k)$. Since the functional posterior is derived from the full observation of a function, this suggests that the relation of conditional entropy $\mathcal{H}(z|\mathcal{D}_\tau^T) \leq \mathcal{H}(z|\mathcal{D}_\tau^C)$. This trait is also empirically supported by Fig. (6), where the functional priors with fewer context points exhibit higher values in the trace of the covariance matrices. As consequence, larger $\alpha$ values impose less constraint on the KL divergence term and results in higher entropy of $p(z|\mathcal{D}_\tau^C; \vartheta)$.

## G EXPERIMENTAL SETUP & IMPLEMENTATION DETAILS

In all experiments, we use Adam (Kingma & Welling, 2013) as the default optimizer for all experiments. Pytorch[5] works as the toolkit to program and run experiments.

The implementation of vanilla NPs/CNPs/ANPs follows the repository[6] and related work (Garnelo et al., 2018a;b; Kim et al., 2019) in DeepMind. The implementation of ML-NPs is based on that in (Foong et al., 2020) except that the convolution modules are removed for the fair comparison since the inference objective is our research focus. To enable researchers to implement our developed method in studies, we leave the anonymous Github link here: `https://anonymous.4open.science/r/SI_NPs-C832`, where we provide an example of SI-NPs. The full code implementations of our method will be released in the final version.

### G.1 META LEARNING DATASETS

**Synthetic Regression.** We use the Gaussian process simulator to generate different stochastic functions. Three types of kernels are used to formulate diverse Gaussian processes. This is the same as that in (Lee et al., 2020; Kawano et al., 2020).

For each task, we generate $x$ from the uniform distribution $U[-2.0, 2.0]$ and the mean function is zero. The number of context points is drawn from $n \sim U[3, 47]$ and the number of target points is $n + m$ with $m \sim U[3, 50-n]$. The following kernel functions specify different Gaussian processes in this paper.

---

[5] `https://pytorch.org/`
[6] `https://github.com/deepmind/neural-processes`

- Matern $-\frac{5}{2}$ kernel:

$$k(x, x') = s^2 \left( 1 + \frac{\sqrt{5}d}{l} + \frac{5d^2}{3l^2} \right) \exp \left( \frac{-\sqrt{5}d}{l} \right)$$

  with $d = 4|x - x'|$, $s \sim U[0.1, 1.0]$ and $l \sim U[0.1, 0.6]$;

- RBF kernel:

$$k(x, x') = s^2 \exp \left( -\frac{(x - x')^2}{2l^2} \right)$$

  with $s \sim U[0.1, 1.0]$ and $l \sim U[0.1, 0.6]$

- Periodic kernel:

$$k(x, x') = s^2 \exp \left( \frac{-2 \sin^2(\frac{\pi ||x - x'||^2}{p})}{l^2} \right)$$

  with $s \sim U[0.1, 1.0]$, $l \sim U[0.1, 0.6]$ and $p \sim U[0.1, 0.5]$

**Image Datasets.** Benchmark image datasets include MNIST (Bottou et al., 1994), FMNIST (Xiao et al., 2017), CIFAR10 (Krizhevsky et al., 2009) and SVHN (Sermanet et al., 2012). In meta training and testing, we randomly select the number of context pixels $n$ for each sampled batch of images ($n \sim U[1, 784]$ in MNIST/FMNIST and $n \sim U[1, 1023]$ in CIFAR10/SVHN). For pixel values, they are transformed to normalized Tensors via pytorch package. All other set-ups are the same as in (Garnelo et al., 2018a;b; Kim et al., 2019).

**Sim2Real Dataset.** This is a part of additional experiments. We retain the preprocessing and meta training set-up in (Gordon et al., 2019). The meta training datasets include the Lotka-Volterra simulation samples and Predator-Prey's real-world dataset. All datasets are normalized before the meta training process. Please refer to (Gordon et al., 2019) for more details.

## G.2 NEURAL ARCHITECTURES & OPTIMIZATIONS & EVALUATION SET-UP

**Synthetic Regression.** In terms of neural architectures, we use the same setup as that in (Gordon et al., 2019; Lee et al., 2020) for all baselines. The dimension of latent variables is 128. The `Encoder` is a two hidden layer neural network with 128 neuron units for each layer. The `Decoder` is a one hidden layer neural network with 128 neuron units. The optimizer's learning rate is $5e - 4$. For all methods, we sample 100 tasks as one batch to train in each iteration, and the number of iteration steps in meta training is 100000. In meta training, the numbers of Monte Carlo samples for ML-NPs and SI-NPs are 16, while those in meta testing are 32.

**Image Completion.** The setup is the same with that in (Garnelo et al., 2018a) and works for all NPs variants. As default, we set the dimension of latent variables $z$ as 128 for all baselines. For all baselines, the `Encoder` is constituted with three hidden layers (128 neuron units each). The `Decoder` has five hidden layers (128 neuron units each) as well. The learning rate for the optimizer is $5e - 4$. The training batch size for all images is 4 and we meta train the model until convergence (the maximum epoch number for MNIST/FMNIST is 100, and that for CIFAR10/SVHN is 200, and early stop is used when it reaches convergence). In meta training, the numbers of Monte Carlo samples are 16 for ML-NPs and 8 for SI-NPs (We find that 8 Monte Carlo samples are enough for SI-NPs to obtain competitive performance and increasing the number of particles will require more computations but the performance improvement is minor). In meta testing, the Monte Carlo sample numbers are 32 for latent variable models.

**Sim2Real.** We use the same setup as that in (Gordon et al., 2019; Lee et al., 2020) for all baselines. The neural architectures are the same as the Synthetic regression settings, except the dimension of output variables changes to 2. For other settings, please refer to (Gordon et al., 2019) for more details.

As in (Garnelo et al., 2018a), for the `Decoder` in all models, we use the modified standard deviation variable $\hat{\sigma_i} = 0.1 + 0.9 * \sigma_i$ for the output distribution $p(y_i|x_i, z; \vartheta)$ in all benchmarks, where $\sigma_i = \text{MLP}_\vartheta(x_i, z_i)$.

**Evaluation Set-up of Datasets.** For Table (1)/(2) in meta testing, we keep the same set-up as (Foong et al., 2020; Gordon et al., 2019; Lee et al., 2020; Kawano et al., 2020), which randomly select the number of context points in each batch and average the testing results of all batches as the final result. The range of the number of context points can be found in the above subsection.

# H    MORE EXPERIMENTAL RESULTS

## H.1    SI-NPS WITH A LEARNABLE PROPOSAL DISTRIBUTION

We have investigated the use of a learnable proposal distribution in SI-NPs. In this case, an additional proposal distribution $q_\eta(z) = \mathcal{N}(z; \mu_\eta(\mathcal{D}_\tau^T), \Sigma_\eta(\mathcal{D}_\tau^T))$ is introduced to sample the latent variables. The neural architecture of such a proposal distribution is the same as the approximate posterior in vanilla NPs, but the role is entirely distinguished.

Unfortunately, we find that it is difficult to stabilize performance even though coefficients of the generative log-likelihood $\ln p(\mathcal{D}_\tau^T | z^{(b)}; \vartheta)$, the functional prior log-likelihood $\ln p(z^{(b)} | \mathcal{D}_\tau^C; \vartheta)$ and the proposal likelihood $\ln q_\eta(z^{(b)} | \mathcal{D}_\tau^T)$ are tuned in a lot of trials. Since there is no convergence solution when assigning the same weights to minimize Eq. (31) and the negative of Eq. (12) with a shared optimizer, we do not report the result here.

## H.2    EVALUATION WITH MORE MONTE CARLO PARTICLES

In the main paper, evaluating the exact likelihood $\mathcal{L}(\vartheta) = \ln \left[ \int p(\mathcal{D}_\tau^T | z; \vartheta) p(z | \mathcal{D}_\tau^C; \vartheta) dz \right]$ is intractable for the studied latent variable models, so we report the evaluation results by setting $B$ the number of the Monte Carlo particles in Eq. (39) fixed.

$$\mathcal{L}_{\text{MC}}(\vartheta; B) = \ln \left[ \frac{1}{B} \sum_{b=1}^{B} \exp \left( \ln p(\mathcal{D}_\tau^T | z^{(b)}; \vartheta) \right) \right] \quad \text{with} \quad z^{(b)} \sim p(z | \mathcal{D}_\tau^C; \vartheta) \tag{39}$$

**Empirical Explanation.** We set $B = 32$ in evaluation based on the empirical observations because the evaluated likelihood for all experiments nearly reaches the convergence or does not significantly increase. To see this point, we give the illustration of the evaluation results by varying the number of Monte Carlo samples and analyzing the test log-likelihoods in image completion tasks. As displayed in Fig. (8), SI-NPs show performance improvement with more particles in FMNIST/CIFAR10/SVHN, and $B = 32$ is sufficient enough in evaluation[7]. For CNPs, the performance does not change with more particles since the functional prior is collapsed. While for NPs, the performance goes far worse than importance-weighted ones and does not significantly improve with more particles. So we only show the ML-NPs and SI-NPs for better visual comparison.

**Theoretical Explanation.** Note that ML-NPs and SI-NPs are importance weighted methods, the number of Monte Carlo samples matters. It is theoretically proved in Theorem 1 in the paper importance weighted autoencoders (Burda et al., 2016) that $\mathcal{L}_{\text{MC}}(\vartheta; B_1) \geq \mathcal{L}_{\text{MC}}(\vartheta; B_2)$ with $B_1 \geq B_2$. Our developed SI-NPs can be viewed as the conditional version of importance weighted autoencoders, which explains the empirical observations in Fig. (8). However, when the prior is collapsed to a deterministic embedding, we do not expect this effect.

## H.3    INFLUENCE OF DIMENSIONS OF LATENT VARIABLES

It is unrealistic to explore all hyper-parameters' influence: e.g., learning rates, numbers of layers, dimensions of each layer, types of activation function, batch size in training, dimensions of latent variables, the Cartesian of these hyper-parameters causes dimension explosion, and the required time of running experiments can be more than one year with limited GPUs. So we keep most of the set-up of the above hyper-parameters the same as that in original papers of CNP/NP/ANPs.

---

[7]Even though in FMNIST, more performance gain is observed using SI-NPs with more particles, $B = 32$ is sufficient and fair enough for all benchmarks.

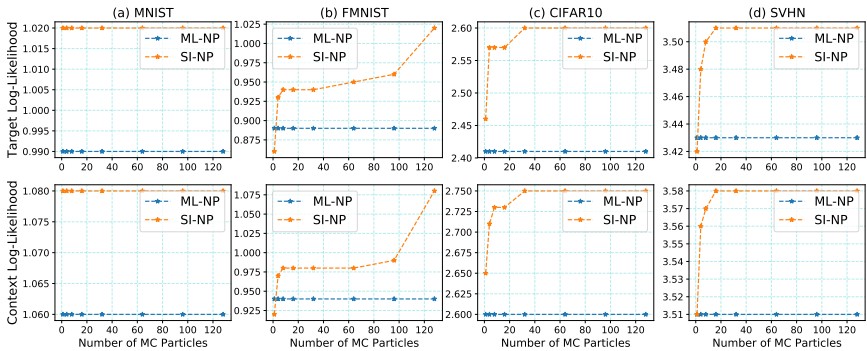

Figure 8: Evaluation of Image Completion with Varying Number of Monte Carlo Particles. The $x$-axis records the number of Monte Carlo particles $B$ in the set $\{1, 4, 8, 16, 32, 64, 96, 128\}$. The first row is to report the log-likelihood of the target points, while the second row is for the log-likelihood of the context points.

In this subsection, we study the influence of dimensions of latent variables on SI-NPs. Considering that the meta training process is time expensive, we report the result on FMNIST image completion in Table (6). It can be observed that this factor rarely influences test performance.

Table 6: Test average log-likelihoods with reported standard deviations for image completion in FMNIST (5 runs). We test the performance of different optimization objectives in both context data points and target data points. We use 32 Monte Carlo samples from the functional prior to evaluate the average log-likelihoods.

| # | dim_lat $= 32$ | | dim_lat $= 64$ | | dim_lat $= 128$ | | dim_lat $= 256$ | |
|---|---|---|---|---|---|---|---|---|
| | context | target | context | target | context | target | context | target |
| $\mathcal{L}_{\text{SI-NP}}$ (ours) | $0.98_{\pm 0.006}$ | $0.95_{\pm 0.004}$ | $0.98_{\pm 0.004}$ | $0.93_{\pm 0.005}$ | $0.98_{\pm 0.004}$ | $0.94_{\pm 0.005}$ | $0.98_{\pm 0.004}$ | $0.93_{\pm 0.005}$ |

## H.4 SIM2REAL EXPERIMENTAL RESULTS

Following that in (Gordon et al., 2019; Lee et al., 2020), We conduct experiments in Lotka-Volterra dynamical systems. The transition dataset is collected in the mentioned simulator for meta training. The testing scenarios include Lotka-Volterra simulators and a Predator-Prey's real-world dataset Hudson's Baye hare-lynx.

The meta testing results are reported in Table (7). It can be seen that SI-NPs can achieve the best performance in Lotka-Volterra and Predator-Prey datasets. Since the Predator-Prey dataset is out of the meta training distribution, the log-likelihoods are significantly lower than the Lotka-Volterra ones.

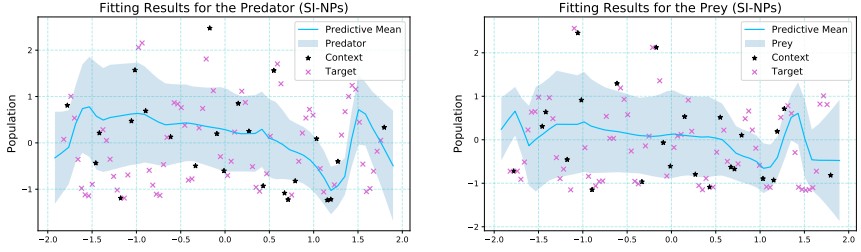

Figure 9: From the Left to the Right are population fitting results with $\pm 1$ standard deviation in the predator and prey datasets with the meta trained SI-NPs. The $x$-axis corresponds to normalized years from 1845 to 1935 in order.

Meanwhile, we plot the predicted predator and prey population evolution in Fig. (9). As illustrated, meta trained SI-NPs can roughly characterize the trends and they can capture critical turning points.

Table 7: Test average log-likelihoods of target data points with reported standard deviations for Sim2Real (4 runs). We test the performance of different optimization objectives in both context data points and target data points. For each run, we randomly sample 1000 functions as tasks to evaluate.

| # | Sim (Lotka-Volterra) | Real(Predator-Prey) |
|---|---|---|
| $\mathcal{L}_{\text{NP}}$ | $-0.457_{\pm 0.015}$ | $-3.275_{\pm 0.507}$ |
| $\mathcal{L}_{\text{CNP}}$ | $-0.273_{\pm 0.026}$ | $-3.012_{\pm 0.034}$ |
| $\mathcal{L}_{\text{ML-NP}}$ | $-1.381_{\pm 0.196}$ | $-2.969_{\pm 0.165}$ |
| $\mathcal{L}_{\text{SI-NP}}$ (ours) | $-0.240_{\pm 0.052}$ | $-2.934_{\pm 0.33}$ |

## H.5 AUGMENTING SI-NPs WITH ATTENTION NETWORKS

Theoretically, we can combine different optimization objectives of NPs with various structural inductive biases. Take the attention inductive bias as an example; we augment the vanilla SI-NP with attention networks the same as in (Kim et al., 2019) and compare the augmented one with other augmented baselines. To enable fair comparison, we also augment other baselines with attention networks. We apply the modification to all methods, and this operation results in ANP (Kim et al., 2019), ML-ANP and SI-ANPs.

**Neural Architectures.** We retain all neural architectures the same as in Appendix (G.2), except for adding the deterministic path to obtain a local deterministic variable (Kim et al., 2019). The deterministic path is built with a cross attention encoder. Due to memory restriction, one head is used to obtain a 128 dimensional local variable, and the neural network set-up for query/key/value is the same as (Kim et al., 2019)[8].

**Optimization.** We retain the optimization step the same as those in Appendix (G.2).

### H.5.1 SYNTHETIC REGRESSION

In Table (8), it can be seen SI-ANPs outperform ANPs in all kernel cases. SI-ANPs show a slight advantage over ML-ANPs in Marten and RBF kernels. We also observe anomaly results of ANPs in the Periodic kernel case, which illustrates that the use of attention might deteriorate the performance.

Table 8: Test average log-likelihoods of target data points with reported standard deviations for 1-dimensional Gaussian process dataset with various kernels (5 runs). For each run, we randomly sample 1000 functions as tasks to evaluate. All NP models are augmented with attention networks. Settings are same as in the main paper.

| # | Matern $-\frac{5}{2}$ | RBF | Periodic |
|---|---|---|---|
| $\mathcal{L}_{\text{ANP}}$ (Kim et al., 2019) | $0.98_{\pm 0.021}$ | $1.06_{\pm 0.020}$ | $-1.346_{\pm 0.102}$ |
| $\mathcal{L}_{\text{ML-ANP}}$ (Foong et al., 2020) | $0.985_{\pm 0.019}$ | $1.062_{\pm 0.019}$ | $0.574_{\pm 0.023}$ |
| $\mathcal{L}_{\text{SI-ANP}}$ (ours) | $0.995_{\pm 0.017}$ | $1.071_{\pm 0.017}$ | $0.56_{\pm 0.024}$ |

### H.5.2 IMAGE COMPLETION

In Table (9), we report the results in image completion. We notice that the SI-ANP significantly beats other models in FMNIST/SVHN/CIFAR10 and is comparable with the ML-ANP in MNIST. Meanwhile, the performance gap between vanilla NPs and SI-NPs in Table (2) is quite huge, while this situation is alleviated after adding attention modules. All of these indicate that incorporating structural inductive biases in complicated tasks is also necessary. Combining SI-NPs with more powerful structural inductive biases is the top choice for boosting performance.

---

[8]https://github.com/deepmind/neural-processes

Table 9: Test average log-likelihoods of target data points with reported standard deviations for image completion in MNIST/FMNIST/SVHN/CIFAR10 (4 runs). All NP models are augmented with attention networks. Same as in the main paper, we test the performance of different optimization objectives in target data points. We use 32 Monte Carlo samples from the functional prior to evaluate the average log-likelihoods.

| # | MNIST | FMNIST | SVHN | CIFAR10 |
|---|---|---|---|---|
| $\mathcal{L}_{\text{ANP}}$ (Kim et al., 2019) | $1.173_{\pm 0.008}$ | $1.101_{\pm 0.01}$ | $4.011_{\pm 0.005}$ | $3.605_{\pm 0.016}$ |
| $\mathcal{L}_{\text{ML-ANP}}$ (Foong et al., 2020) | $1.216_{\pm 0.003}$ | $1.172_{\pm 0.009}$ | $4.017_{\pm 0.002}$ | $3.545_{\pm 0.01}$ |
| $\mathcal{L}_{\text{SI-ANP}}$ (ours) | $1.212_{\pm 0.004}$ | $\mathbf{1.174}_{\pm 0.005}$ | $\mathbf{4.040}_{\pm 0.002}$ | $\mathbf{3.710}_{\pm 0.028}$ |

### H.5.3 SIM2REAL

In Table (10), we report the results with the attention module augmentation. As observed, ANPs, ML-ANPs, and SI-ANPs exhibit comparable performance in Lotka-Volterra simulation. As for the Predatory-Prey testing results, the conclusion is similar to that in Table (7) without attention augmentations.

Table 10: Test average log-likelihoods of target data points with reported standard deviations for Sim2Real (4 runs). We test the performance of different optimization objectives augmented by attention inductive bias (Kim et al., 2019) in both context data points and target data points. For each run, we randomly sample 1000 functions as tasks to evaluate.

| # | Sim (Lotka-Volterra) | Real(Predator-Prey) |
|---|---|---|
| $\mathcal{L}_{\text{ANP}}$ | $2.211_{\pm 0.017}$ | $-3.174_{\pm 0.121}$ |
| $\mathcal{L}_{\text{ML-ANP}}$ | $2.203_{\pm 0.042}$ | $-3.624_{\pm 0.152}$ |
| $\mathcal{L}_{\text{SI-ANP}}$ (ours) | $2.203_{\pm 0.026}$ | $-2.822_{\pm 0.316}$ |

In this case, we guess the local deterministic embedding from the attention network plays a more critical role than the global latent variable in fitting these two datasets. The visualization of fitting Predator-Prey samples with SI-ANPs is given in Fig. (10). The real-world samples are well fitted with well quantified standard deviations. We notice that the measured predictive mean square errors and the negative log-likelihoods of the target data points in the Predator-Prey testing dataset are comparable using SI-NPs and SI-ANPs. However, the fitting result of the context data points are distinguished a lot: the mean square error of SI-NPs in 4 runs is $0.285\pm0.007$ and that of SI-ANPs is $0.029\pm0.004$. This implies that the attention network in the real-world dataset focuses more on the context point fitting while the global latent variable is for the general trend characterization.

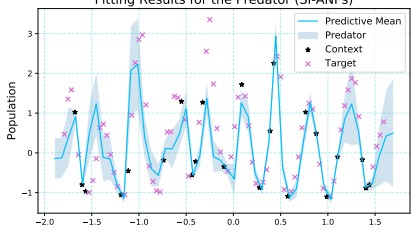 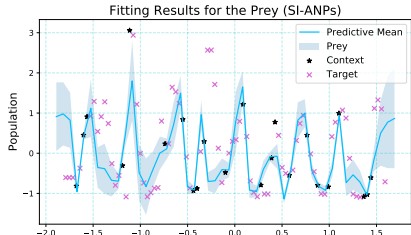

Figure 10: From the Left to the Right are population fitting results with $\pm1$ standard deviation in the predator and prey datasets with the meta trained SI-ANPs. The $x$-axis corresponds to normalized years from 1845 to 1935 in order.

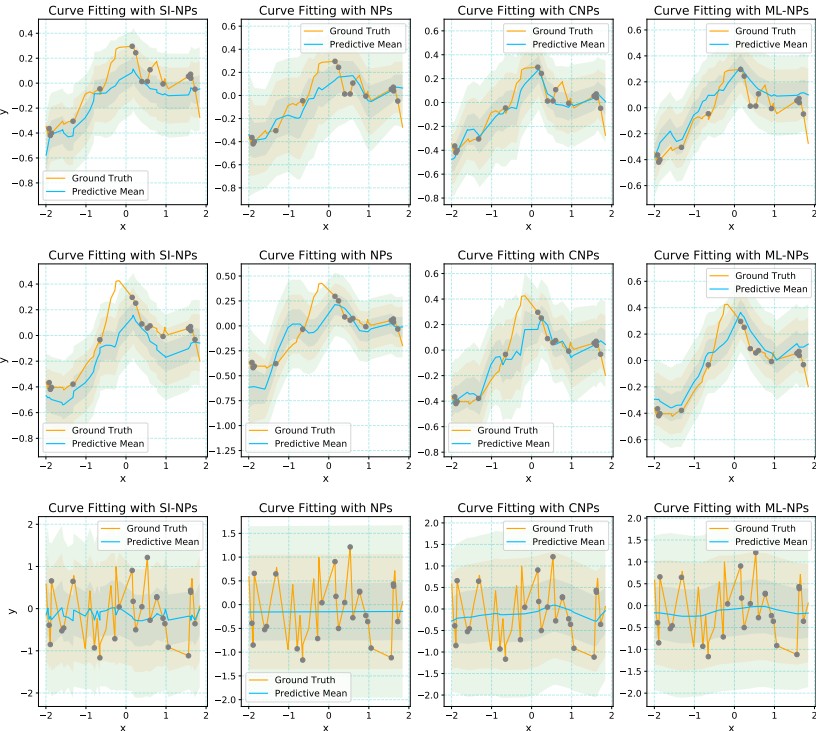

Figure 11: Examples of curve fitting in all kernel cases. From the up to the down in rows are respectively Marten, RBF, and Periodic cases.

## H.6 ADDITIONAL VISUALIZATIONS

### H.6.1 MORE SYNTHETIC REGRESSION RELATED RESULTS

We include more visualized results with the meta-trained models for Gaussian process datasets in Fig. (11). An illustrated, periodic kernel cases are most challenging, and SI-NPs can learn faint but crucial fluctuation signals from mean functions, while vanilla NPs fail to capture them. For other cases, the behaviors of these models are similar in characterizing trends: vanilla NPs tend to show higher variance in Marten kernel cases. SI-NPs and ML-NPs are comparable in Marten and RBF kernel cases. CNPs seem to best match the context points in both cases.

In Fig.s (12)/(13)/(14), we respectively plot the curve fitting results in all kernel cases when all latent variable models are augmented by attention networks. In Marten and RBF cases, all attention augmented models well fit data points. Nevertheless, in RBF cases, SI-NPs can better capture data point-dependent deviations. In Periodic cases, it can be observed that SI-ANPs can precisely capture fluctuations and quantify more convincing uncertainty. Sometimes, ANPs underestimate the uncertainty while ML-ANPs fail to show data point distinguished uncertainty.

### H.6.2 MORE IMAGE COMPLETION RELATED RESULTS

In Fig. (15), the scale of computed KL divergence values in vanilla NPs is positively correlated with the semantic complexity. Also, note that the approximate functional prior in vanilla NPs seldom collapses, but it has a theoretical bias away from the optimal functional prior according to **Remark** (1). We can also find with more context points, the approximate prior is closer to the approximate posterior, so the value decreases accordingly.

In Fig. (16), we sample a collection of images from MNIST/FMNIST/SVHN/CIFAR10 to visualize the completed results. We can find that without structural inductive biases, SI-NPs can reasonably complete the images in MNIST/FMNIST/SVHN and exhibit the uncertainty from partial observations. The learned functional prior, such as that in SVHN, can generate images that are different from

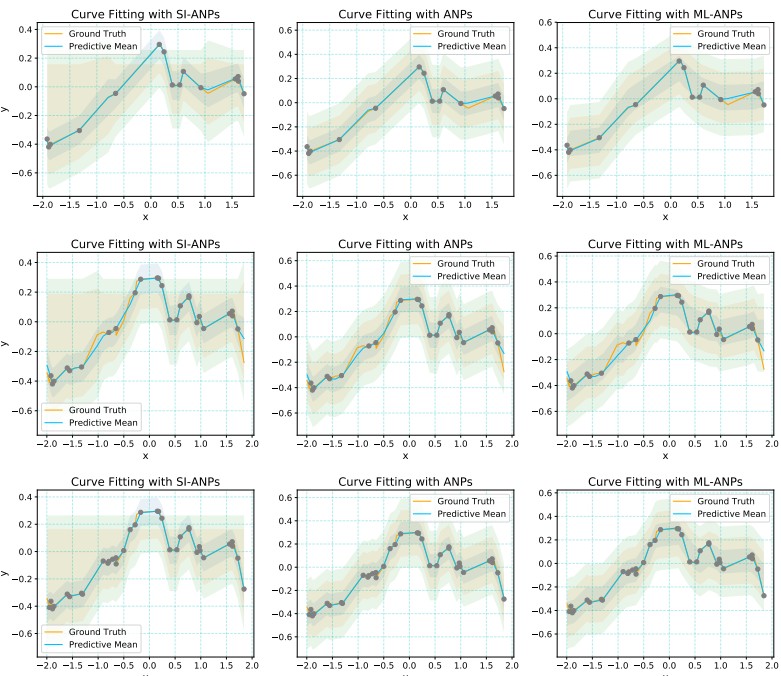

Figure 12: Examples of Curve Fitting in Matern Kernel Cases.

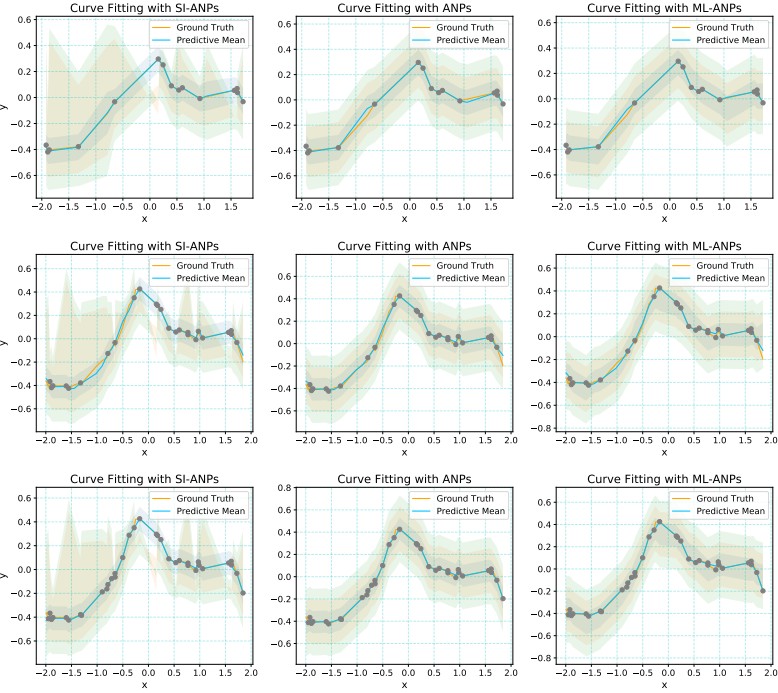

Figure 13: Examples of Curve Fitting in RBF Kernel Cases.

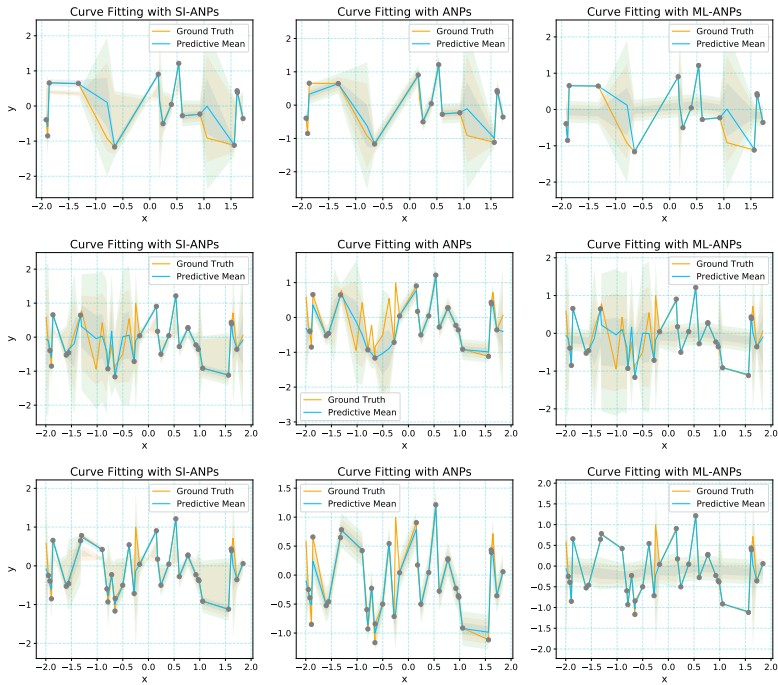

Figure 14: Examples of Curve Fitting in Periodic Kernel Cases.

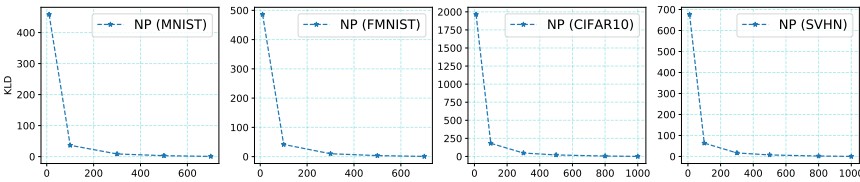

Figure 15: Evaluation of KL Divergence Terms in Vanilla NPs. In meta testing, we still vary the number of context points in image datasets. For NPs, the KL divergence value $D_{KL}\left[q_\phi(z) \,\|\, q_\phi(z|\mathcal{D}_\tau^C)\right]$ is computed.

the ground truth due to partial observation. As for CIFAR10, it has more complicated semantics and is challenging for SI-NPs with only MLPs in neural architectures.

For cases when SI-NPs are augmented by attention neural networks (Kim et al., 2019), we show more generated examples with learned SI-ANPs in Fig. (17)/(18). As can be seen, the completed results are not blurred and have a high quality. Notably, the quantified variances are decreased with the increase in the context points, which shows excellent asymptotic behavior in this domain.

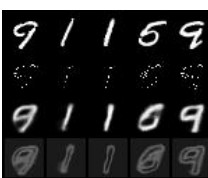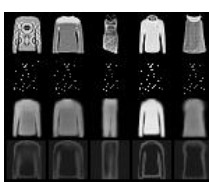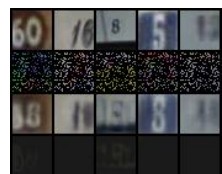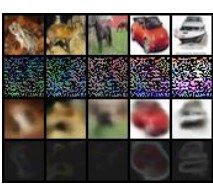

Figure 16: Examples of Image Completion Results using SI-NPs. From top to bottom in rows are original images, context points, means, and variances of completed images.

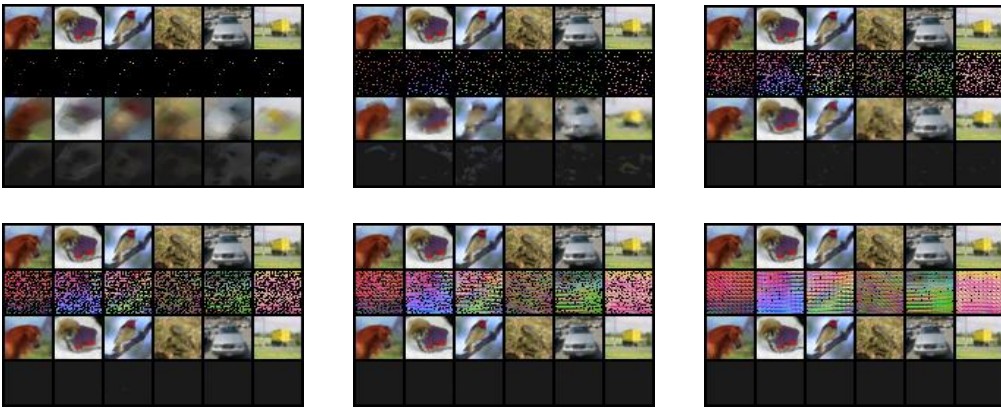

Figure 17: Examples of CIFAR10 Image Completion Results using SI-ANPs. From left to right and top to bottom are cases with 10, 100, 300, 500, and 800 randomly selected pixels as the context points.

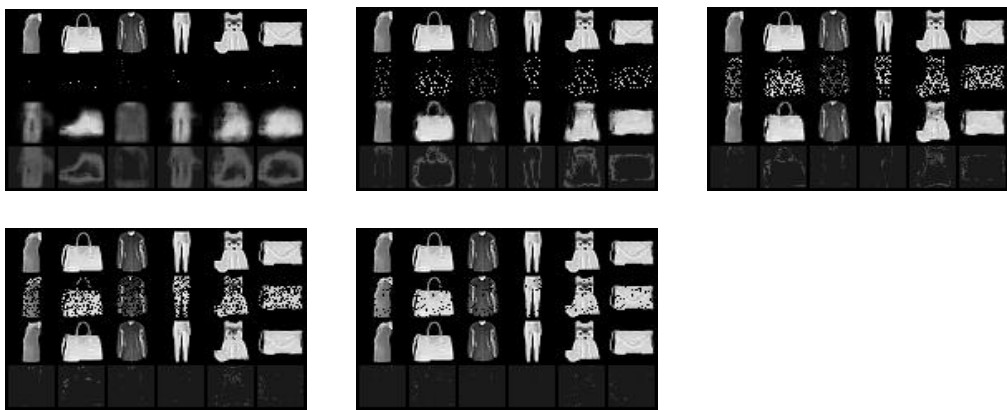

Figure 18: Examples of FMNIST Image Completion Results using SI-ANPs. From left to right and top to bottom are cases with 10, 100, 300, 500, and 700 randomly selected pixels as the context points.

# I    COMPUTATION TOOLS

In this project, we use NVIDIA 1080-TiGPUs to finish all experiments.

