# OpenReview forum: "Bridge the Inference Gaps of Neural Processes via Expectation Maximization"
_ICLR.cc/2023/Conference — ICLR 2023 poster_

### Official Review · Reviewer_BUNX · 2022-10-20

**Confidence:** 4
**Correctness:** 2
**Technical Novelty And Significance:** 3
**Empirical Novelty And Significance:** 2
**Recommendation:** 3

**Clarity, Quality, Novelty And Reproducibility:**

The question of the validity of standard NP's variational approach studied by the paper is a well-known issue in NP research and definitely requires further investigation. Thus, I was happy to see a contribution tackling this. Unfortunately, the submission lacks clarity, so I find it hard to judge the validity of the proposed objective function:

Sec. 3:

- I'm not fully convinced by Remark 1. I agree that, following Eq. (19), $L_{\mathrm{NP}}$ is no lower bound to the log marginal predictive likelihood, and that this can decrease the quality of the solutions of the variational optimization problem. However, also optimizing the (intractable) $\mathcal L_{\mathrm{ELBO}}$ does not "guarantee finding optimal or locally optimal solutions", unless the posterior approximation is perfect. Thus, I would not regard "the consistent regularizer [as] the source of the inference suboptimality of vanilla NPs", or at least not as the only source. I encourage the authors to clarify this.
- Figure 1: I know that this figure is the same as in the original paper by Garnelo et al. Nevertheless, I think it does not exactly fit the descriptions in the paper, which can lead to confusion. Indeed, the central issue studied by the paper is that $p(z|D^C; \vartheta)$ "[is] unknown", which is technically not correct if we start from Fig. 1, where $p(z|D^C; \vartheta)$ is defined to be part of the model definition (the arrow points from the context set to the latent variable). I think it makes more sense to invert this arrow (cf., e.g., [1,2]), as (i) this fits the data generating process better (context and target data come from the same distribution), and (ii) this fits the setting in the paper better, as it discusses intractability of $p(z|D^C; \vartheta)$. Then, the discussions in Sec. 3 should be valid. However, I have doubts about the validity of Sec. 4, cf. below.

Sec. 4: my main concern with this submission is that the main Sec. 4 is quite hard to follow which makes it hard to judge the validity of the approach:

- While the paper motivates in Sec. 3 why NP's objective requires further investigation, it does not provide enough intuition for the proposed EM-like approach, i.p., because central parts are moved to App. E. I would propose to restructure this section and merge parts of App. E into the main part of the paper. Also, I encourage the authors to improve formulations like "In Eq. (6), we take the step by replacing the approximate posterior with the last time updated $p(z | \mathcal D_\tau^T; \vartheta_k)$ in Algorithm (1)" (cf. text above Eq. (10)), which are barely understandable.
- From a technical point of view, it is unclear to me why Eq. (12), i.e., the novel objective the paper proposes, should be tractable. Indeed, in Sec. 3, the it is explained that standard NPs have to introduce the "consistent regularizer" (Eq. (7)) because "the functional prior is unknown". In Eq. (12), this functional prior also appears, why should it be tractable here? Furthermore, it is used as the proposal distribution, why can we now assume to be able to sample from the functional prior?
- Furthermore, I encourage the authors to provide a more extensive discussion of the differences and commonalities of (i) the standard NP objective, (ii) the MC-based NP objective, (iii) the proposed objective, and why one should be superior compared to the other. I.p., the MC-based objective does not suffer from standard NP's intractability issues of the exact ELBO -- why should the proposed method work better?


Unfortunately, also the empirical evaluation does not convince me that the proposed approach is interesting enough to be published in its current form. The proposed method does not improve upon the state of the art on most of the presented experiments. Indeed, the only statistically significant improvements are observed on image completion tasks, where the deterministic CNP approach performs second best. This shows that the image completion tasks are not suitable to properly judge epistemic uncertainty estimation (as *deterministic* methods can solve it well), and, thus, these tasks are not really relevant for judging the effectiveness of the proposed method, which aims to improve upon the training objective of *latent variable* NP models. I also encourage the authors to elaborate on the following concerns w.r.t. the experimental evaluation:

- Did you optimize hyperparameters? It seems like you just fixed all hyperparameters without tuning? This is definitely not a fair approach, as NP architecture's performance heavily depends on hyperparameters. Please provide results after tuning at least the learning rate and the latent dimension (at least on the synthetic data sets), cf., e.g., [2].
- Could you provide the exact formulae you used to compute the metrics given in the tables and asymptotic performance plots? What are the context sizes you used in the tables?
- $B=32$ samples to evaluate the marginal predictive likelihood could be too little to obtain reasonable performance estimates, cf., e.g., [3]. Could you provide results showing that $B=32$ is enough, i.e., that the performance estimates do not improve anymore with more samples?
- Please report confidence intervals instead of standard deviations.
- Could you add confidence intervals in Fig. 5?

Minor comments/typos:

- I find the notation w.r.t. the approximate distributions $q_\phi$ confusing. Sometimes it is denoted as $q_\phi(z|D^T)$ and sometimes the abbreviation $q_\phi(z)$ is used. One could think about not using the abbreviation at all, which would also make the distinction between $q_\phi(z|D^T)$ and $q_\phi(z|D^C)$ more explicit.
- p.1/3rd paragraph: $[x_*.y_*]$ -> $[x_*,y_*]$.
- Eq. (18c): The approximate posterior should not have a dependence on $\vartheta$.
- Eq. (19): The distribution $q_\phi(z)$ in the subscript of the expectation should be conditioned on $D^T$.
- Above Eq. (7): What do the authors mean with "the posterior in the exact ELBO [is] unknown"? As far as I understand, the only intractable term in the ELBO is the functional prior?
- The objective Eq. (9) was actually already introduced in [1].

[1] Gordon et al., "Meta-Learning Probabilistic Inference for Prediction", ICLR 2019

[2] Volpp et al., "Bayesian Context Aggregation for Neural Processes", ICLR 2020

[3] Grosse et al., "Sandwiching the marginal likelihood using bidirectional Monte Carlo", Arxiv 2015


**Strength And Weaknesses:**

The paper tackles an important question in NP research, namely the validity of approximating what the paper calls the "functional prior" with the set encoder distribution. Unfortunately, the presentation of the approach lacks clarity, which makes it hard to judge its validity. Also, the experimental results are inconclusive. Please cf. my comments below.


**Summary Of The Paper:**

The paper studies Neural Process (NP)-based meta regression. The work is motived by the well-known observation that the exact ELBO of NPs is intractable, which requires an approximation that destroys the guarantee that the resulting NP-objective is a lower bound to the log marginal predictive likelihood. The paper proposes a novel objective function for NPs, inspired by an Expectation Maximization (EM)-like approach, which is shown to offer an improvement guarantee. The approach is validated on several synthetic and image-completion experiments.


**Summary Of The Review:**

The paper tackles an important question in NP-based research. Unfortunately, the presentation lacks clarity and the experimental results are inconclusive. In it's current form, I judge this paper not ready for publication, but encourage the authors to provide an improved version, incorporating my suggestions above.

---

> ### Author Response · Authors · 2022-11-14
> **Response to Reviewer BUNX (Part I)**
>
> We sincerely thank **# Reviewer BUNX** for the constructive comments.
> The followings focus on answers to questions and clarifications on misunderstandings.
>
> ***1. Approach Presentation Lacks Clarify & Validity Issues.***
>
> *We updated the manuscript to clarify the concerns raised by the reviewer. These efforts include:*:
>
> - Prepared an anonymous link to the **manuscript slides/PPT in the first paragraph on Page15**  for **# Reviewer BUNX** to better understand
> - **Polished the first paragraph of Section 4** for a quick overview
> - Rewrite the sentence by “*Following the Algorithm (1), we take the E-step #1 by replacing the approximate posterior in Eq. (6) with the last time …*” on **Page5**.
> - **Additional explanations for the Method part**: It consists of three steps in proof: *i) define the meta learning surrogate function in Eq. (22) as required by EM algorithms, ii) execute the Expectation step and the Maximization step, and bound the results with the likelihood of meta learning dataset in Eq. (23), iii) loop the EM steps and obtain the iteration inequalities to show the improvement guarantee.* Meanwhile, **Fig. (2)** visually shows these steps.
>
> For ***the validity issues***, *we leave detailed Remark/Propositions proofs step by step* in **Section E**. Meanwhile, *we proofread this part multiple times, and the validity of the method was also examined* by **# Reviewer sw5r** and **# Reviewer HNS9**.
>
> ***2. Experimental Results are Inconclusive.***
>
> *Our investigation is through the lens of optimization objectives and pays more attention to 1) checking whether the EM framework alleviates the inference suboptimality of vanilla NPs and 2) understanding the randomness of the functional prior*. Please refer to **the Global Response III. Clarifications on Contributions** to see conclusive results.
>
> ***3. Understanding Remark1 and Fig.1.***
>
> Thanks for the comment.
>
> - For **Remark1**, *there exist other suboptimality sources even for the exact ELBO*, e.g. approximation gaps in Table5. But in **Remark1**, we stress ***the NP objective cannot bound the likelihood and fails to judge the connection to the log-likelihood $\mathcal{L}(\vartheta)$ or optimization direction***. We polished Remark1 by adding, “*Eq. (7) is an invalid variational inference objective*”.
>
> - **Fig.1** shows the *factorization of the generative process for NPs*. *It refers to the generative and not the inference process*. ***To resolve your confusion, we added “show the generative process” on Page1***. ***The functional prior $p(z|\mathcal{D}^{C};\vartheta)$ is part of the model definition, but we need to iteratively learn its optimal parameter because this is unknown***.
>
> ***4. Clarifications on Central Part & Eq12 Tractable Objective.***
>
> Thanks for the comment.
>
> - We need to clarify that **the central part of methods** is **Section4.1 with a high-level method summary/Fig.2/Algorithm1**,  instead of the mentioned **App. E** (Section E are just the step-by-step math proof).
>
> - **Eq(12) is tractable since it is the self-normalized importance sampling result, note that the posterior is not Gaussian and cannot be used to sample for Monte Carlo estimates of Eq(11)**. *You can find this explanation below Prop.2 “Still we cannot optimize…” **on Page6**. Since we model the functional prior as Gaussian in modeling, we can directly sample latent variables from the last time updated distribution.*

---

> > ### Author Response · Authors · 2022-11-14
> > **Response to Reviewer BUNX (Part II)**
> >
> > ***5. Summarize Differences and Commonalities in NP/MC-based NP/SI-NP.***
> >
> > Thanks for the comment.
> > **Please refer to Connections between Different Optimization Objectives and Table 3 on Page15/16**.
> > We also explained *why self-normalized importance-weighted methods work* by adding, *“Since the estimates of importance weights in SI-NPs exploit the target observations and consider the difference in particles…”*.
> >
> > ***6.Should Optimize Hyper-parameters for Fairness.***
> >
> > Thanks for the comment.
> > *We partially disagree with this comment *:
> > >“Did you optimize hyperparameters? It seems like you just fixed all hyperparameters without tuning? This is definitely not a fair approach…”.
> >
> > Note that ***our experimental set-up of hyperparameters, e.g., learning rate, dimension of latent variables, and neural architectures, is mostly the same as the original papers of vanilla CNP/NP/ANPs/BoostrappingNP [1-4] and many other works [5-7]***.
> > You can further double-check our paper and the experimental set-up of these works.
> > ***We retain them, just aiming for a fair comparison.***
> >
> > Meta-training is time-consuming in practice. ***It is unrealistic to explore all hyper-parameters influence: e.g., learning rates, numbers of layers, dimensions of each layer, types of activation function, batch size in training, dimensions of latent variables, the Cartesian of these hyper-parameters causes dimension explosion, and the required time of running experiments (4 baselines x 7 benchmarks x 5 runs) can be more than one year with limited GPUs.***
> >
> > Nevertheless, *we give the example of FMNIST image completion and report the influence of $z$'s dimensionality on SI-NPs*. This factor is not decisive and does not influence the final conclusion.
> >
> > For the evaluation of **Table1/2**, *we follow the same set-up in work [4-7], you can find the updated descriptions “Evaluation Set-up of Datasets. For Table (1)/(2) in meta testing…”* at the beginning of **Page26**.
> >
> >
> > ***7. Result Analysis with More MC Particles.***
> >
> > This is a good question. For SI-NPs, the empirical results with more MC particles can be further improved to a certain level or converge in evaluation when the functional prior is not collapsed. We answer this in **Section. H.2 and Fig.8 on Page27**, where we show *the results with the varying numbers of MC particles {1, 4, 8, 16, 32, 64, 96, 128}. When the number of particles is greater than 32, the evaluation performance reaches either convergence or marginal improvement for all benchmarks.*
> >
> >
> > ***8. Please Report Confidence Intervals instead of Standard Deviations.***
> >
> > Thanks for the comment.
> > ***Reporting the standard deviations is the standard way of evaluation in most of NPs related work [2,4-7].***
> > *We didn’t find valid ways to calculate confidence intervals for Table results.*
> > Especially, you can find ***the standard deviations in Table2 are quite small with the scale 1e-2 or 1e-3, these tiny values are hardly visually shown in Fig.5.***
> >
> > ***9. Other Typos.***
> >
> > Thanks for pointing them out. These typos have been revised already:
> > - We keep the approximate posterior in App. consistent with the main paper now. (using the abbreviation $q_{\phi}(z)$ due to line width restrictions)
> > - Revised the comma typo [x_*,y_*]
> > - Revised the appendice typo Eq. (18c)
> > - The posterior is intractable in the sense of sampling in Monte Carlo estimates
> > - Added one more reference to Eq.9 by adding, “(or VERSA (Gordon et al., 2018)).” on Page4.
> >
> > ***Finally, we hope your questions are well answered, and thanks for your efforts once again.***
> >
> > ***References:***
> >
> > [1] Garnelo, Marta, et al. "Conditional neural processes." International Conference on Machine Learning. PMLR, 2018.
> >
> > [2] Garnelo, Marta, et al. "Neural processes." arXiv preprint arXiv:1807.01622 (2018).
> >
> > [3] Kim, Hyunjik, et al. "Attentive neural processes." arXiv preprint arXiv:1901.05761 (2019).
> >
> > [4] Lee, Juho, et al. "Bootstrapping neural processes." Advances in neural information processing systems 33 (2020): 6606-6615.
> >
> > [5] Gordon, Jonathan, et al. "Convolutional Conditional Neural Processes." International Conference on Learning Representations. 2019.
> >
> > [6] Kawano, Makoto, et al. "Group Equivariant Conditional Neural Processes." International Conference on Learning Representations. 2020.
> >
> > [7] Foong, Andrew, et al. "Meta-learning stationary stochastic process prediction with convolutional neural processes." Advances in Neural Information Processing Systems 33 (2020): 8284-8295.

---

> > > ### Comment · Reviewer_BUNX · 2022-12-02
> > > **Remaining concerns (1/2)**
> > >
> > > Thank you for your updates and the clarifications. Unfortunately, a central point of the submission still remains unclear to me.
> > >
> > > The paper studies a generative model defining a log marginal predictive likelihood of the form
> > >
> > > $$
> > > \log p(\mathcal D^T_\tau | \mathcal D^C_\tau, \vartheta) = \log \int p(\mathcal D^T_\tau | z, \vartheta) p(z | \mathcal D^C_\tau, \vartheta)\mathrm d z
> > > $$
> > >
> > > which is bounded by the “exact ELBO” $\mathcal L_{\mathrm{ELBO}}(\vartheta, \phi)$, i.e.,
> > >
> > > $$
> > > \log p(\mathcal D^T_\tau | \mathcal D^C_\tau, \vartheta) \geq \mathcal L_{\mathrm{ELBO}}(\vartheta, \phi)
> > > $$
> > >
> > > with
> > >
> > > $$
> > > \mathcal L_{\mathrm{ELBO}}(\vartheta, \phi) = \mathbb E_{q_{\phi}(z | \mathcal D^T_\tau)}\left[\log p(\mathcal D^T_\tau | z, \vartheta) + \log p(z | \mathcal D^C_\tau, \vartheta) - \log q_\phi(z | \mathcal D^T_\tau) \right],
> > > $$
> > >
> > > where $q_\phi(z | \mathcal D^T_\tau)$ is typically computed by a set-encoder architecture. As stated in the paper below Eq. (6), for non-empty context sets $\mathcal D^C_\tau \neq \varnothing$, “the functional prior […] in the exact ELBO [is] unknown”, as it involves computing the intractable evidence term $p(\mathcal D^C_\tau | \vartheta)$ in the denominator of Bayes’ theorem:
> > >
> > > $$
> > > p(z | \mathcal D^C_\tau, \vartheta) = \frac{p(\mathcal D^C_\tau | z, \vartheta) p(z | \vartheta) }{p(\mathcal D^C_\tau | \vartheta)}.
> > > $$
> > >
> > > The standard NP architecture makes use of the learned set-encoder $q_\phi(z | \mathcal D^T_\tau)$ (which can be conditioned on arbitrary data sets), to define a surrogate objective
> > >
> > > $$
> > > \mathcal L_{\mathrm{NP}}(\vartheta, \phi) = \mathbb E_{q_{\phi}(z | \mathcal D^T_\tau)}\left[\log p(\mathcal D^T_\tau | z, \vartheta) + \log q_\phi(z | \mathcal D^C_\tau) - \log q_\phi(z | \mathcal D^T_\tau) \right].
> > > $$
> > >
> > > In Remark 1 the authors (correctly) state that this surrogate objective in general does not constitute a lower bound to the log marginal predictive likelihood, i.e.,
> > >
> > > $$
> > > \log p(\mathcal D^T_\tau | \mathcal D^C_\tau; \vartheta) \ngeq \mathcal L_{\mathrm{NP}}(\vartheta, \phi).
> > > $$
> > >
> > > What I find contradictory is that in the author’s answer to my review it is stated that that “the functional prior $p(z | \mathcal D^C_\tau, \vartheta)$ is part of the model definition”, i.e., tractable by design. I emphasize that I do not state that it is invalid to make this assumption. But if this assumption is made, Remark 1 does not apply anymore, as the exact ELBO $\mathcal L_{\mathrm{ELBO}}(\vartheta, \phi)$ is tractable and using the surrogate objective $\mathcal L_{\mathrm{NP}}(\vartheta, \phi)$ is not necessary. Furthermore, as suggested in my initial review, I would also change notation if this assumption is made. For example, I would denote the functional prior by $\tilde p(z | \mathcal D^C_\tau, \vartheta)$, as this does not suggest that it is connected to the model likelihood $p(\mathcal D^C_\tau | z, \vartheta)$ by virtue of Bayes’ theorem as given above. Likewise, I would denote the resulting log marginal predictive likelihood by
> > >
> > > $$
> > > \log \tilde p(\mathcal D^T_\tau | \mathcal D^C_\tau; \vartheta) \equiv \log \int p(\mathcal D^T_\tau | z, \vartheta) \tilde p(z | \mathcal D^C_\tau, \vartheta)\mathrm d z.
> > > $$
> > >
> > > Then,
> > >
> > > $$
> > > \log \tilde p(\mathcal D^T_\tau | \mathcal D^C_\tau, \vartheta) \geq \mathcal L_{\mathrm{ELBO}}(\vartheta, \phi) = \mathbb E_{q_{\phi}(z | \mathcal D^T_\tau)}\left[\log p(\mathcal D^T_\tau | z, \vartheta) + \log \tilde p(z | \mathcal D^C_\tau, \vartheta) - \log q_\phi(z | \mathcal D^T_\tau) \right]
> > > $$
> > >
> > > with — as stated above — $\mathcal L_{\mathrm{ELBO}}(\vartheta, \phi)$ being fully tractable. Thus, assuming a tractable functional prior (as the authors state in the rebuttal) should make the proposed approach obsolete.
> > >
> > > Let us thus again consider the original setting where the functional prior $p(z | \mathcal D^C_\tau, \vartheta)$ is intractable, which is the setting that motivates the proposed objective due to the shortcomings of $\mathcal L_{\mathrm{NP}}(\vartheta, \phi)$ as discussed in Remark 1. The authors propose a new objective of the form
> > >
> > > $$
> > > \mathcal L_{\mathrm{SI-NP}}(\vartheta, \eta_k, \vartheta_k)=\sum_{b=1}^B \hat \omega^{(b)} \left[ \log p(\mathcal D^T_\tau | z^{(b)}, \vartheta) - \log p(z^{(b)} | \mathcal D^C_\tau, \vartheta) \right],
> > > $$
> > >
> > > where the intractable functional prior $p(z | \mathcal D^C_\tau, \vartheta)$ appears both in $\hat \omega^{(b)}$ and as the second term in the square bracket. I do not see why this objective should now be tractable to evaluate?

---

> > > > ### Comment · Reviewer_BUNX · 2022-12-02
> > > > **Remaining concerns (2/2)**
> > > >
> > > > In summary:
> > > >
> > > > - Either the functional prior is tractable, then the exact ELBO $\mathcal L_{\mathrm{ELBO}}$ is tractable and there is no need for the proposed approach.
> > > > - Or the functional prior is intractable, then the standard approach of using $\mathcal L_{\mathrm{NP}}$ as a surrogate ELBO presents problems from an inference perspective (as summarized in Remark 1), but also the objective $\mathcal L_{\mathrm{SI-NP}}$, proposed to alleviate this problem, should be intractable.
> > > >
> > > > I ask the authors again to elaborate on this contradiction.

---

> > > > > ### Author Response · Authors · 2022-12-02
> > > > > **Replies to Concerns**
> > > > >
> > > > > Dear Reviewer # BUNX,
> > > > >
> > > > > We’re very glad to receive your feedback. The following is to address your concerns:
> > > > >
> > > > > ***(1) Concerns on tractable functional prior***
> > > > >
> > > > > In this paper, we are using a tractable functional prior, a Gaussian distribution parameterized with neural networks.
> > > > > In this case, the exact ELBO is also available.
> > > > > However, as summarized in Appendice **Table5**, we point out that optimizing the ELBO suffers **the posterior approximation gap**.
> > > > >
> > > > > Especially, note that ***the exact functional posterior $p(z|D^T;\vartheta)=\frac{p(D^T|z;\vartheta)p(z|D^C;\vartheta)}{\int p(D^T|z;\vartheta)p(z|D^C;\vartheta)dz}$ is a significantly complicated Non-Gaussian distribution, while the Gaussian distribution family works as the default approximate posterior in the VI framework (in most of NP models).
> > > > > This can cause huge approximation gaps in functional posteriors, and the theoretical suboptimality w.r.t. the likelihood $\mathcal{L}(\vartheta)$ cannot be resolved when optimizing the NP models.***
> > > > >
> > > > > In contrast, our SI-NP retains the same neural architecture as the NP and optimizes the model within the Expectation Maximization framework.
> > > > > We also demonstrate the convergence to at least a local optimum with our developed EM method.
> > > > >
> > > > > ***(2) More details on optimizing the importance-weighted objectives***
> > > > >
> > > > > Note that the computation of $\hat{\omega}$ relies on the last time updated prior $p(z^{(b)}|D^C;\vartheta_k)$, which means the $\vartheta_k$ is fixed and not updated.
> > > > > ***This corresponds to the $E$-step in our proposed EM algorithm, and the gradient computation only applies to expression $[ln p(D^T_{\tau}|z^{(b)};\vartheta)-ln p(z^{(b)}|D^C_{\tau};\vartheta)]$ in the bracket Eq (12)*** (Please refer to Fig.2, Pseudo Algorithm, and the descriptions below Eq. (12)).
> > > > >
> > > > > For any questions, feel free to reply to us. And we hope your concerns are well addressed.

---

### Official Review · Reviewer_ki6V · 2022-10-23

**Confidence:** 1
**Correctness:** 3
**Technical Novelty And Significance:** 3
**Empirical Novelty And Significance:** 2
**Recommendation:** 6

**Clarity, Quality, Novelty And Reproducibility:**

I am only superficially aware of NPs, so won't try to judge the novelty, quality, and reproducibility. The paper seems clearly written.

As a minor point, I wish the authors have discussed whether there is any trade-off between using more samples from $q_\eta$ (better estimate of the integral), and loosing gradient signal for update of the $\eta$ parameters. An issue of this kind appeared in the IWAE literature, but perhaps is not a problem here?

**Strength And Weaknesses:**

### strengths
* sensible theoretical motivation
* better empirical results than baselines

### weaknesses
* the empirical improvement is somewhat marginal compared to the oracle (see Table 1)
* increased computational complexity

**Summary Of The Paper:**

This paper investigates a new technique for inference in neural processes (NPs) based on expectation maximisation. The technique is related to what was proposed by Foong et al. (2020)—convolutional NPs—where multiple samples from the latent given task context are sampled to construct the likelihood objective. Unlike Foong et al., the authors here propose to use importance weighting to approximate expectation with respect to the true posterior. This is motivated by author's theoretical derivations which show that the consistency regulariser used in previous work on NPs causes suboptimal inference.

**Summary Of The Review:**

I am not an expert on NPs. AFAICT, the paper is well written, and presents an advancement to the state of knowledge about NPs. If that is correct, I believe it justifies acceptance, even if some of the empirical gains are marginal, and the computational cost is somewhat high.

---

> ### Author Response · Authors · 2022-11-14
> **Response to Reviewer ki6V**
>
> We sincerely thank **# Reviewer ki6V** for the constructive comments.
> The followings focus on answers to questions.
>
> ***1. Empirical Improvements & Computational Cost.***
>
> Thanks for the question.
> - *We keep the neural architecture of models all the same and evaluate the results from multiple aspects, including NLLs, the prior collapse effect, and the complexity of function families.*
> The empirical improvements and connections with remarks/propositions are summarized in the **III. Clarifications on Contributions of the Global Response**.
>
> - Since ML-NPs and SI-NPs are importance weighted methods, both require more samples in training.
> *But with a bit more computational cost, SI-NPs seldom encounter prior collapse and can generate more diverse functions from the latent space (which is the goal of deep generative models).*
>
>
> ***2. “...advancement to the state of knowledge about NPs. If that is correct, I believe it justifies acceptance”.***
>
> Thanks for the comment.
> - *For Remarks, Propositions, and Math Equations, we have provided all proofs and derivations step by step.*
> We double-check all proofs and equations multiple times throughout the submission.
> Meanwhile, **# Reviewer sw5r** and **# Reviewer HNS9** have examined on these math parts.
> - *As far as we know, this is the first work concerning inference suboptimality and theoretical analysis of optimization objectives for NPs.*
>
> ***3. Connections with IWAE.***
>
> Thanks for the question.
> *In the vanilla IWAE, the prior is fixed, and the weights for all particles are equal.*
> In comparison, *ours is a self-normalized importance-weighted method*, and *we did not observe the fading gradient signals with limited numbers of particles*. A more detailed analysis is attached in **Section H.2 on Page26**.
>
> *We hope your questions are well answered, and thanks for your efforts once again.*

---

### Official Review · Reviewer_HNS9 · 2022-11-03

**Confidence:** 4
**Correctness:** 4
**Technical Novelty And Significance:** 3
**Empirical Novelty And Significance:** 2
**Recommendation:** 6

**Clarity, Quality, Novelty And Reproducibility:**

The paper is nicely written and provides an easy to follow introduction into the relevant neural process literature and backgrounds.

Section 4 could benefit from a little reorganization by potentially first discussing what strategy will be used and then introducing that step by step instead of starting with an algorithm box and introducing the relevant theory one page down.

In terms of quality and content I enjoyed the paper and think it has a lot of value for the community since it focuses on the narrow idea that the inference strategy for a popular model class matters and digs into that empirically and theoretically, while proposing a relevant and principled avenue towards improvement.

**Strength And Weaknesses:**

Strengths:
- studying the NP objective thoroughly and sidestepping some of the previous simplifying assumptions previously made is interesting and relevant here
- the authors propose an interesting strategy using importance sampling to estimate expectations over p(z|D_t) and utilize that in a variational EM scheme
- the results show quantitative evidence that this strategy outperforms the more commonly used training strategies

Weakness:
- While the authors bring up uncertainty at some point, the results shown here do not suggest that the ucnertainty estimates are dramatically improved
- the function prior still basically collapses to a delta function, which feels like a pathological issue in the setup and is not quite prevented here merely by the inference scheme
- inside the variational EM scheme there are still dramatically simplified assumptions present such as setting variational families to the prior etc., this is still relatively dissatisfactory when considering that this prior tends to collapse
- in the GP example, the NP models proposed here outperform the baseline models, but are still dramatically far off from the GP.  This suggests that the NP model class does not really appear to be a suitable model as suggested here for this modeling task and surprises me. I would expect that a suitably large neural model with enough data should at least get predictive means similarly right as a GP, it is not clear to me this is the case here for the entire model class irrespective of inference strategy. If so, how useful is it to compare to GPs , really?


**Summary Of The Paper:**

The authors study neural processes from an inference perspective, drilling into questions about how well the NP family really works as a function of the objective. In the course of this analysis, they propose a self normalized impottance sampling scheme to estimate the expectation over the posterior over the data latent variable z in p(z|D_t) and use that to obtain a  tighter estimate of the training objective.
They link this procedure to variational expectation maximization and show how iterative optimization of this objective leads to better NP models.

As an aside, the authors study the resulting "prior collapse" in their function prior p(z|D_c) as a function of training objective and link it to performance and diversity in function space.

**Summary Of The Review:**

This work focuses on providing more depth into our understanding of neural processes from an inference point of view and suggest a modification which improves upon the commonly used strategies in a principled way.
While a lot of issues remain with the model class, the clean approach used here and modest empirical gains suggest that this work should be interesting to the community working on neural processes and adds useful knowledge.

Given those I would lean towards recommending this paper for acceptance, even though I remain interested in unsolved topics surrounding modeling aspects of NPs which remain unadressed here but may also affect uncertainty and similar aspects the authors touch upon.

---

> ### Author Response · Authors · 2022-11-14
> **Response to Reviewer HNS9**
>
> We sincerely thank **# Reviewer HNS9** for the constructive comments.
> The followings focus on answers to questions.
>
> ***1.Improvement on Uncertainty Estimates & Prior Collapse.***
>
> In our experiments, *simply analyzing the output Log-likelihoods is not sufficient for deep generative models like NPs*.
> *We evaluate the results from multiple aspects, including Log-likelihoods, the prior collapse effect, and the complexity of function families*.
> And these empirical observations and connections with our remarks/propositions are summarized in the **III. Clarifications on Contributions of the Global Response**.
>
> ***2. Proposal Distributions in the Variational EM.***
>
> This is a good question.
> In practice, the reweighted wake-sleep algorithm has no unified optimization objective [1]. **On Page15**, we answer this question by "*The reason why we set the update of learnable proposal distribution an optional choice is that we find it difficult to balance the optimization of Eq. (12) and Eq. (31). This is non-trivial and assigning equal weights in optimization results in unstable performance in experiments (Please refer to **Section (H.1)** for more discussion.). ….*"
>
> *References:*
>
> [1] Le, Tuan Anh, et al. "Revisiting reweighted wake-sleep for models with stochastic control flow." Uncertainty in Artificial Intelligence. PMLR, 2020.
>
> ***3. Comparisons with Oracle on Synthetic Regression.***
>
> Thanks for this question, and we updated the answer **on Page15**.
> The GP oracle is computed in the form
> $\ln p(y_{1:n+m}\vert y_{1:n},x_{1:n+m})$, where the predictive distribution is mainly with a non-diagonal covariance matrix (suggesting $y_{n+m}$ not independent in statistics).
> In comparison, the log-likelihood of NPs family is computed in the form
> $\sum_{i=1}^{n+m}\ln p(y_i\vert [x_i,z];\vartheta)$, where $y_{1:n+m}$ is conditional independent w.r.t. the global latent variable $z$.
> Such a difference in computations causes the mentioned gap.
> *However, with ***a large neural model***, such as the integration of attention networks in **Section H.5**, the quantified performance gap of GP Oracles and the log-likelihood of NPs family is well reduced*.
>
> ***4. A Little Reorganization of Sec. 4.***
>
> Thanks for this suggestion and *we’ve taken your advice and polished the manuscript*.
> Besides, we attach the anonymous link to the slides of this work **on Page15**.
>
> *We hope your questions are well answered, and thanks for your efforts once again.*

---

### Official Review · Reviewer_sw5r · 2022-11-03

**Confidence:** 3
**Correctness:** 3
**Technical Novelty And Significance:** 4
**Empirical Novelty And Significance:** 3
**Recommendation:** 8

**Clarity, Quality, Novelty And Reproducibility:**

This paper is neatly written, theoretically sound and novel in terms of both providing insightful analysis of the problem and proposing novel framework.

The experiments are well documented, and should be reproducible given the enough hyper-parameter details are covered in the paper. It would be great if the code could be shared along with camera ready version.



**Strength And Weaknesses:**

Strength:

The paper did thorough analysis on an important issue of inference sub-optimality of NPs, and introduced the novel self-normalized importance weighted neural processes model to address the issue. The paper is well written with comprehensive derivation in the appendix.

Weaknesses:

This is a great paper, but the reviewer has some minor questions hoping that the author(s) could help to explain or address:

1. Is there any proof of Proposition 2? It is unclear to the reviewer why it leads to an improvement *guarantee* w.r.t. the log-likelihood.

2. In 4.2, it mentions that "skip E-step #2 to avoid unstable optimisation in empirical results." Does it mean that using a learnable functional prior as the proposal is the key to avoid unstable optimisation? Is there any case that the author(s) would suggest to do E-step #2 to update proposal distribution in wake phase update?

3. In 5.1 Table 1, for the functions sampled from GP with Matern - 5/2 kernel, the performances of SI-NP and ML-NP are of no statistically significant difference, but for ones with RBF or periodic kernels, SI-NP is much better than ML-NP. Is there any intuition or insights on this?

4. In 5.2 Table 2, it seems that the improvements of SI-NP vs CNP or ML-NP is marginal on image complementation tasks. Would you expect a larger improvement with higher number of MC samples?

5. The Sim2Real experiments in the appendix seems very interesting. It would be nice to have it in the main text and with some insightful discussion.

**Summary Of The Paper:**

The paper addresses the inference sub-optimality issue of neural processes. To achieve competitive performance of NPs, the paper proposes a surrogate objective of the target log-likelihood in a meta learning setup, and introduces a tractable way to optimize the surrogate objective through variational expectation maximisation. The author(s) further validates the method with image completion tasks on MNIST, FashionMNIST, SVHN and CIFAR10.

**Summary Of The Review:**

This paper is solid for publication in ICLR.

---

> ### Author Response · Authors · 2022-11-14
> **Response to Reviewer sw5r**
>
> We sincerely thank **# Reviewer sw5r** for the constructive comments.
> The followings focus on answers to questions.
>
> ***1. Proof of Proposition 2 Regarding Improvement Guarantee.***
>
> Thanks for your question.
> Due to the page limit, we leave the detailed proof of Proposition2 in **E.1 on Page20**.
> It consists of three steps in proof: i) *define the meta learning surrogate function in Eq. (22) as required by EM algorithms*, ii) *execute the Expectation step and the Maximization step, and bound the results with the likelihood of meta learning dataset in Eq. (23)*, iii) *loop the EM steps and obtain the iteration inequalities to show the improvement guarantee*.
> **Fig. (2)** is to visually show these steps.
>
> ***2. Necessity of Functional Prior as the Proposal Distribution.***
>
> This is a good question.
> In practice, the reweighted wake-sleep algorithm has no unified optimization objective [1], so the answer is “not mandatory to use the prior as the proposal”. **On Page15**, we answer this question by "*The reason why we set the update of learnable proposal distribution an optional choice is that we find it difficult to balance the optimization of Eq. (12) and Eq. (31)….*"
> We also attached the code example of SI-NPs' implementation in **Section.G on Page 24**, allowing additional trials to update the independent proposal available in our code.
>
> *References*:
>
> [1] Le, Tuan Anh, et al. "Revisiting reweighted wake-sleep for models with stochastic control flow." Uncertainty in Artificial Intelligence. PMLR, 2020.
>
> ***3. More Discussions on Synthetic Regression.***
>
> Thanks for this question. We notice that in comparison to RBF and Periodic kernels, *the scale parameter of Matern kernels also varies with kernel values, bringing more variations in realizations. Maybe a global latent variable is difficult to handle in this case, and SI-NPs and ML-NPs encounter a similar performance bottleneck.*
>
> ***4. Result Analysis with More MC Particles.***
>
> This is a good question.
> *For SI-NPs, the empirical results with more MC particles can be further improved to a certain level or converge in evaluation when the functional prior is not collapsed*.
> We answer this in **Section. H.2 and Fig.8 on Page27**, where we show the results with the varying numbers of MC particles {1, 4, 8, 16, 32, 64, 96, 128}.
> *When the number of particles is greater than 32, the evaluation performance reaches either convergence or marginal improvement for all benchmarks.*
>
> ***5. Moving Additional Results to the Main Paper.***
>
> Thanks for this kind suggestion. Due to the page limit, we leave Sim2Real experimental results in supplementary materials.
>
> *We hope your questions are well answered, and thanks for your efforts once again.*

---

### Author Response · Authors · 2022-11-13
**Global Response**

We sincerely thank all the Reviewers for their precious comments and constructive suggestions.

### **I. Summary of Positive Comments**
These include: ***(1) the well/neatly-written paper with comprehensive derivations*** by **# Reviewers sw5r/HNS9/ki6V** ***, (2) thorough, insightful analysis on NPs inference suboptimality/studied the inference strategy with a lot of values for the community*** by **# Reviewers sw5r/HNS9**, ***(3) a novel/technically sound model with an empirical and a proven theoretical improvement*** by **# Reviewers sw5r/HNS9**, and ***(4) advancement to the NPs knowledge*** by **# Reviewer ki6V**.

### **II. Primary Changes in the Manuscript**
The updated manuscript is to ***answer questions***, ***polish descriptions***, and ***clarify misunderstandings*** for some comments. The changed contents in the manuscript are marked in red. These include:

1. ***Added an anonymous link to the manuscript slides/PPT*** in the ***first paragraph on Page15*** for **# Reviewer BUNX** and other readers to better understand **the framework of the method and theoretical/empirical discoveries**.

2. ***Explained the effect of more MC particles in the empirical evaluation of log-likelihoods*** and why we set the fixed number as reasonable in **Section H.2**.

3. ***Polished the first paragraph of Section 4***.
4. ***Added theoretical explanation*** of why self-normalized importance-weighted methods work better than equally importance-weighted methods on **Page15**.
5. ***Added experimental results*** to show the influence of ***the latent variable’s dimension*** on the performance in **Section H.3**.
6. ***Answered the question of NPs performance evaluation*** on GPs and difference with oracle evaluations on **Page15**.

### **III. Clarifications on Contributions**
Instead of simply pursuing SOTA performance, we take more interest in the ***connections between remarks/propositions and empirical observations***:

- ***inference suboptimality inside vanilla NP in Remark1 and improvement guarantee in Proposition1/2*** → empirically, SI-NPs show significant performance improvement over vanilla NPs

- ***CNP’s equivalence in Proposition3 and prior collapse*** → empirically, SI-NPs show marginal improvement but with less chance of prior collapse, suggesting the potential of generating more diverse functions from the latent space.

- ***functional prior and semantics complexity*** → empirically, datasets with less rich semantics, e.g. MNIST, the deterministic functional representation may be sufficient for representing function families.

*Finally, we hope our extra efforts, including anonymous paper slides, polished content, and additional empirical results, have well resolved your concerns. Thanks to all reviewers for collaborating with us to improve the manuscript.*

---

### Author Response · Authors · 2022-11-27
**Any Replies from the Reviewers**

Dear All Program Committee Members,

The rebuttal and the revised manuscript were posted two weeks ago, and we are still waiting for a reply to the rebuttal.
Since the time is quite limited to address the questions/concerns according to the guideline, ***can the reviewers give some updates on reviews recently?***

Finally, we thank your efforts in this phase.

---

### Author Response · Authors · 2022-12-04
**Significance of our Proposed Method (ELBO and Importance Weighted Objective?)**

Dear All,

***Based on Reviewer # BUNX’s response to our Rebuttal, it seems most of the raised concerns were well addressed except the motivation of our proposed optimization objective.***

According to ***Reviewer # BUNX’s second round summary and deduction*** when a parameterized Gaussian works as the functional prior (***the basic set-up/assumptions in nearly all NP work***):

>
.. **the exact ELBO $\mathcal{L}_{ELBO}$ is tractable, so there is no need for the proposed approach (our Self-Normalized Importance Objective).**
>
With regard to the above deduction, we need to point out ***some misunderstandings about Variational Inference and Importance Weighted Methods***.

***Clarifications on VI/IW methods:***
---------------

(1) Reviewer # BUNX ignores ***the inherent posterior approximation gap (together with the amortization gap, but not our primary focus in this work)*** and ***maximizing $\mathcal{L}(\vartheta)$ is the ultimate goal of NP models***.
Remember the VI decomposition: $L(\vartheta)=L_{ELBO}(\vartheta;\phi)+E_{q_{\phi}(z)}[q_{\phi}(z)||p(z|D^C;\vartheta)]>L_{ELBO}(\vartheta;\phi)$ with ***the prior $p(z|D^C;\vartheta)$ a Gaussian***.

However, ***the exact functional posterior $p(z|D^T)=\frac{p(D^T|z;\vartheta)p(z|D^C;\vartheta)}{\int p(D^T|z;\vartheta)p(z|D^C;\vartheta)dz}$ is a significantly complicated Non-Gaussian distribution, while the Gaussian distribution family works as the default approximate posterior in the VI framework (in most of NP models)***.

This can cause huge approximation gaps in functional posteriors, and ***the theoretical suboptimality cannot be resolved when optimizing ELBOs.***

(2) In contrast, our SI-NP retains ***the same neural architecture (Gaussian priors) as the NP and uses the same evaluation method***. The benefit of using the Expectation Maximization framework is ***to directly bound the $\mathcal{L}(\vartheta_k)\leq\mathcal{L}(\vartheta;\vartheta_k)\leq\mathcal{L}(\vartheta_{k+1})$ with $\mathcal{L}(\vartheta;\vartheta_k)$ the meta surrogate function***.
We also demonstrate ***the convergence to at least a local optimum*** with our developed EM method.

***Conclusion:***
---------------
(1) In comparison to exact ELBO, ***our self-normalized importance weighted objective does not suffer the posterior approximation gap and directly optimizes $\mathcal{L}(\vartheta)$ in a surrogate way***.

(2) Also, ***the importance-weighted methods have a theoretically tighter likelihood bound based on the proof of Importance Weighted Autoencoder work*** (So it is not appropriate to draw the conclusion that **if we can obtain the ELBO, we do not need to develop new methods like importance-weighted ones.** Otherwise, all importance-weighted methods are meaningless.).

Finally, we thank reviewers for the second round of replies and hope all concerns from reviewers are addressed in the second round of rebuttal.

---

> ### Comment · Reviewer_BUNX · 2022-12-05
> **Answer**
>
> Thanks for your answer! I still think that the submission is not ready for acceptance at ICLR because central claims made in the paper apparently do not hold. This was also confirmed by the authors in their response during the rebuttal. Let me again repeat my concerns.
>
> It is completely obvious to me and to anyone else slightly familiar with VI methods that the tightness of the KL bound and, thus, the efficiency of the optimization procedure, depends on the expressiveness of the variational distribution, and that a simple Gaussian task posterior approximations will typically incur a signification approximation gap in the NP setting. I also agree with the authors that it is interesting and necessary to study improved objectives for NP training to reduce this gap and that the well-known (e.g., from the VAE literature) importance-weighted approach proposed in the submission might by an interesting method to study in the NP setting. Thus, I not at all “ignore the inherent posterior approximation gap”.
>
> As I stated repeatedly, my problem with the submission is that it claims to identify another source of inference suboptimality that is unique to NPs. Indeed, the authors claim that one of the main contributions of the paper is to identify that “most previous work ignores the reason why the vanilla NP suffers the performance bottleneck” and that the submission helps “understanding the inference suboptimality of vanilla NPs”.  Indeed, the submission “claim[s] that the consistent regularizer is the source of inference suboptimality of vanilla NPs”. My discussion with the authors during the rebuttal phase shows that this is not the case in the setting studied in the submission (as the exact ELBO is tractable), invalidating one of the main contributions of the paper. In fact, it appears to me that the authors are now trying to sell the standard VI approximation gap as the problem they tackle in their paper. I emphasize again that this can be an interesting contribution, but **I strongly believe that the paper needs to be reworked to make it clear to the reader that it does not provide a solution to the problem of the exact ELBO of NPs not being tractable, but that it tackles the standard VI problem of the exact ELBO being a loose bound to the log marginal predictive likelihood if the posterior approximation is poor**.

---

> > ### Author Response · Authors · 2022-12-05
> > **Happy to Recieve Replies and Welcome Discussions**
> >
> > Hi Reviewer # BUNX,
> >
> > Thanks for your further comments, and we welcome heated discussions.
> >
> > (1) To avoid disputes, we propose rephrasing the sentence
> > >"The consistent regularizer is the source of the inference suboptimality of vanilla NPs"
> >
> > by
> > >"The consistent regularizer is the source of the inference suboptimality of vanilla NPs since it does not have the improvement guarantee w.r.t. $L(\vartheta)$ as illustrated in Remark1."
> >
> > Since NP is a surrogate ELBO, we cannot find the direct bound relationship between $L_{NP}$ and $L(\vartheta)$ in both tractable/intractable functional cases.
> >
> > (2) Throughout our latest manuscript, *we circumvent disputes on optimizing either exact or intractable ELBO (we focus on new strategies to optimize in any case)*.
> > ***When the NPs' functional prior is intractable, we abandon the standard VI or approximate ELBO and directly construct/optimize the surrogate function w.r.t. $L(\vartheta)$ with likelihood bounds.***
> >
> > ***This operation is the same as the related work $L_{ML-NP}(\vartheta)$ [1].*** In [1], it also points out ***$L_{NP}$ is an invalid ELBO, and directly using a Gaussian parameterized functional prior can theoretically result in better performance***.
> >
> > (3) ***Since ours $L_{SI-NP}(\vartheta)$ and $L_{ML-NP}(\vartheta)$ are importance-weighted NPs [1], we do not tackle the standard VI problem***. Instead, ***we use Self-Normalized Importance Sampling to directly approximate $L_{\vartheta}$ rather than fixing exact/intractable ELBO***.
> >
> > In brief, ***our proposed solution is independent of traditional VI for posterior inference, which is regardless of ELBO***. Feel free to reply to us ***whether you are satisfied with the above rephrase and whether the clarification is clear enough. If not, please inform us where to revise, and we will be happy to do so.*** Thanks.
> >
> > **Reference:**
> >
> > [1] Foong, Andrew, et al. "Meta-learning stationary stochastic process prediction with convolutional neural processes." Advances in Neural Information Processing Systems 33 (2020): 8284-8295.

---

> > > ### Comment · Reviewer_BUNX · 2022-12-12
> > > **Thank you for the answer**
> > >
> > > Dear authors,
> > >
> > > thank you for your answer!
> > >
> > > I decided to keep my score because I still believe that Sec. 3.1/Remark 1 is not relevant/applicable to the paper's setting. I thus leave it up to the area chair to decide how much weight to put on my concern.
> > >
> > > -----
> > >
> > > > we welcome heated discussions
> > >
> > > Please excuse that my reply sounds a bit more "heated" than I actually intended on a second read -- I really appreciate that the authors engaged in a thorough discussion!

---

### Decision · Program_Chairs · 2023-01-20

**Decision:**

Accept: poster

**Justification For Why Not Higher Score:**

While the insights of the paper seem quite useful to the NP community, it's a little niche for an oral.  The reviewers seemed to think that the empirical results might suggest it's not a clear win, but nonetheless most thought the paper should be accepted.

**Justification For Why Not Lower Score:**

The majority voted for an accept, and in the reviews it seemed clear that reviewers found this an interesting, novel and sophisticated technical contribution that would be useful to share with the community.

**Metareview: Summary, Strengths And Weaknesses:**

Neural processes are a relatively recent method for modeling distributions over functions using neural networks.  In this paper, the authors argue that standard inference in neural processes is suboptimal and leads to underfitting.  The authors analyze this issue and propose an alternative objective along with an importance weighting and EM-based algorithm to optimize it.  In experiments, the authors show qualitatively that their method fits data well and that this alternative view outperforms existing NP approaches on a variety of benchmarks.

The review scores were mixed (8, 3, 6, 6), but there was a strong correspondence between what reviewers thought were the strengths and weaknesses of the paper.   Specifically, most reviewers found the paper well motivated, the setting novel and technically interesting and correct.  Most of the reviewers seemed to find that the observations and innovations of the work would be quite interesting to the community (e.g. "it has a lot of value for the community").  The main weakness highlighted by the reviewers was with regard to the empirical evaluation.  Specifically, they seemed to find the improvements over the state of the art somewhat marginal.  One reviewer found the description of the method lacking clarity.  The difference in the reviewer scores seems to reflect their opinion of how important the empirical results are.

In summary, the paper is novel and insightful, methodologically interesting but empirically it's not clear that it's substantially better than the state-of-the-art.  It seems like the paper is complete (i.e. doesn't really require substantial additional work to deliver the message).  Given that the majority of reviewers believed it's a novel and useful contribution to the community, the recommendation is to accept the paper.  Not every paper needs to achieve SOTA to be a useful contribution to the collective research and understanding of a topic.

**Note From Pc:**

if the above contains the word "oral" or "spotlight" please see: "oral" presentation means -> notable-top-5% and "spotlight" means -> notable-top-25%. As stated in our emails, we are disassociating presentation type from AC recommendations